



# Aerosol Optical Properties Measurement using the Orbiting High Spectral Resolution Lidar onboard DQ-1 Satellite: Retrieval and Validation

Chenxing Zha[1], Lingbing Bu[1], Zhi Li[1,2], Qin Wang[3], Ahmad Mubarak[1], Pasindu Liyanage[1], Jiqiao Liu[4], Weibiao Chen[4]

[1] School of Atmospheric Physics, Nanjing University of Information Science and Technology, Nanjing, 210044, China
[2] Nanjing Movelaser Technology Co., Ltd, Nanjing, 210044, China
[3] Tianjin Meteorological Radar Research and Test Centre, Tianjin, 300061, China
[4] Key Laboratory of Space Laser Communication and Detection Technology, Shanghai Institute of Optics and Fine Mechanics, Chinese Academy of Sciences, Shanghai, 201800, China

*Correspondence to*: Lingbing Bu (001779@nuist.edu.cn), Qin Wang (1826262365@163.com)

**Abstract.**

The atmospheric environment monitoring satellite DQ-1 was launched in April 2022, which consists of a spaceborne High Spectral Resolution Lidar (HSRL) system. This new system enables the accurate measurements of global aerosol optical properties, which can be used in Geo-scientific community after the Cloud-Aerosol Lidar and Infrared Pathfinder Satellite Observation (CALIPSO) retirement. Developing a suitable retrieval algorithm and validating retrieved results are prominently needed. This research demonstrates a retrieval algorithm for aerosol optical properties using the DQ-1 HSRL system. This method has retrieved the aerosol depolarization ratio, backscatter coefficient, extinction coefficient, and optical depth. For validation purposes, we compared retrieved results with those obtained through CALIPSO. The results have shown a continuous profile alignment between the two datasets, with DQ-1 describing an improved signal-to-noise ratio of approximately 10 dB. Optical property profiles from NASA Micro Pulse Lidar NETwork (MPLNET) stations were selected for validation with the DQ-1 measurements, resulting in a relative error of 25%. Between June 2022 and December 2022, aerosol optical depth measurements using the DQ-1 satellite and the AErosol RObotic NETwork (AERONET) were correlated and yielded a value of $R^2$ 0.803. We use the DQ-1 dataset to initially investigate the transport processes of the Saharan dust and the South Atlantic volcanic ash. These validations and applications show that the DQ-1 HSRL system can accurately measure global aerosols and holds significant prospects for earth science applications.

**Keywords**: Aerosol Optical Parameters; Spaceborne High Spectral Resolution Lidar; Validation.

## 1 Introduction

Aerosols are tiny solid and liquid particles suspended in the atmosphere, an atmospheric aerosol particle typically ranges from $10^{-3}$ to $10^{-2}$ µm in diameter. The earth's radiation balance is influenced by aerosol's capability to scatter and absorb radiation.





Similarly, as cloud condensation nuclei, aerosols influence cloud formation and affect global climate change (Kaufman et al., 2002). The aerosol optical properties can be used to study the above scientific problems. Lidar, as an active remote sensing instrument, can obtain aerosol optical parameters at a high spatial and temporal resolution. Based on the various observation approaches, lidar can be divided into ground-based lidar, airborne lidar and spaceborne lidar. Typical examples of ground-

based Lidar include the European Aerosol Research LIdar NETwork (EARLINET) (Pappalardo et al., 2014; Guibert et al., 2005), the Micro-Pulsed Lidar (MPL) network (Welton et al., 2001), The Asian Dust and Aerosol Lidar Observation Network (AD-Net) (Nishizawa et al., 2016), etc. There are several advantages of ground-based lidar: easy maintenance of the instrument, stable observation, more accurate results and long-term stable observation of specific areas. Additionally, comparing and validating the datasets between spaceborne and airborne lidar is quite helpful.

The disadvantage of ground-based observation includes limited spatial coverage capability, which makes it impossible to carry out large-scale continuous observation. The airborne lidar system enables extensive and continuous observations over a wide range, thereby compensating for the limitations associated with ground-based systems. Typical examples of airborne Lidar include the NASA Langley airborne High Spectral Resolution Lidar (HSRL-1) and the Langley airborne High Spectral Resolution Lidar - Generation 2 (HSRL-2). The two airborne lidar experiments were conducted to observe aerosol optical

properties quantitatively and to investigate the impact of aerosols on radiation, clouds, and air quality (Knobelspiesse et al., 2011; Burton et al., 2012a; Burton et al., 2012b). The German Aerospace Center (DLR) has also developed an airborne HSRL system to measure aerosol optical properties and types (Esselborn et al., 2008; Groß et al., 2013). The Shanghai Institute of Optics and Fine Mechanics (SIOM) of the Chinese Academy of Sciences, in collaboration with Nanjing University of Information Science and Technology (NUIST), Zhejiang University (ZJU), and other institutions, has conducted observational

experiments at two distinct geographical locations, Dunhuang and Shanhaiguan. The aerosol optical parameters in these two regions were obtained and validated using multi-source data. An analysis was conducted on the sources, sinks, and types of aerosols in the local area (Wang et al., 2020; Xu et al., 2020; Zhu et al., 2021; Juxin et al., 2023; Changzhe et al., 2023). The airborne lidar system addresses the shortcomings of ground-based observations. Nevertheless, the system's observations are limited by factors such as flight paths and meteorological conditions to prevent prolonged data collection.

Global information about aerosol optical parameters is vital to familiarizing with aerosol sources and sinks. This global information values to track aerosol particle dispersion pathways and compensates for the limitations of ground-based and airborne observations. Spaceborne lidar systems operating in orbit have the capability to acquire global aerosol optical parameters (Qin et al., 2016; Huang et al., 2008). With space technology's advancement, several spaceborne lidar systems have been developed. The Cloud-Aerosol Lidar and Infrared Pathfinder Satellite Observation (CALIPSO), developed by NASA, is

the most representative spaceborne lidar satellite. Since its launch in 2006, it has been fully verified by comparing its dataset with other multi-source datasets (Bibi et al., 2015; Wang et al., 2016; Mcgill et al., 2007). Investigation and discrimination of clouds and aerosols, optical properties, types, and microphysical characteristics of the aerosol were performed. An advanced-level retrieval algorithm was developed, to present an outstanding contribution to research on the optical properties and spatiotemporal distribution of aerosols globally (Getzewich et al., 2018; Vaughan et al., 2019; Winker et al., 2010; Chiang et



al., 2011; Liu et al., 2019b). Due to fuel consumption, CALIPSO was retired in August 2023, a well-established and developed new-generation spaceborne Lidar is needed to replace CALIPSO for global aerosol observation. The utilization of filters by the high spectral resolution lidar has filtered the Mie scattering at different echoes, introducing a novel lidar equation. This approach exempts the assumptions made by the traditional lidar retrieval algorithms (Hair et al., 2001). NASA has developed the Cloud-Aerosol Transport System (CATS) as a low-cost payload for the International Space Station (ISS). The system

design incorporates a high spectral resolution lidar and elastic backscattering lidar. It has unveiled the characteristics of aerosols and clouds and their interactions. Similarly, it has conducted in-depth scientific observations in certain areas (Xiong et al., 2023; Proestakis et al., 2019; Yorks et al., 2016). Furthermore, under the leadership of NASA, the Atmosphere Observing System (AOS) international program is dedicated to developing a spaceborne high spectral resolution lidar system designed to operate in a polar orbit. This system aims to unveil ice and water's vertical motion within clouds and provide direct retrieval

algorithms based on high spectral optical systems. The launch of this satellite is planned for the near future (Cornut et al., 2023). ESA and JAXA collaborated on the development of Earth Cloud, Aerosol and Radiation Explorer (EarthCARE), equipped with an ATmospheric LIDar (ATLID). The primary objective of this mission is to observe and characterize clouds and aerosols, as well as to measure the infrared radiation emitted from the Earth's surface and the solar radiation reflected from the atmosphere. This satellite is anticipated to be launched (Reverdy et al., 2015; Wehr et al., 2023). It is worth noting that the

above new spaceborne lidar satellites are in the planning stage and have not actually been launched. The previous observing satellites have stopped working, and now global aerosol data, based on spaceborne lidar, is facing a gap.

   China launched the atmospheric environment monitoring satellite DQ-1 from the Taiyuan Satellite Launch Center on April 16, 2022. The main payload is the Aerosol and Carbon Detection Lidar (ACDL), which includes a dual-polarization HSRL system based on an iodine vapor filter (Liu et al., 2019a; Zheng et al., 2020; Dong et al., 2019). Prior to the launch of DQ-1, an

airborne scaling system for ACDL was developed, deployed, and tested at two sites: Shanhaiguan and Dunhuang. The results ensure the feasibility of the scaling system and verify the accuracy of the observations (Wang et al., 2020; Xu et al., 2020; Zhu et al., 2021; Juxin et al., 2023; Changzhe et al., 2023). The successful operation of the airborne scaling system indicates accurate design and observation of DQ-1 system, and provides a foundation for spaceborne retrieval algorithm. After the above operation, The DQ-1 satellite was launched into a 705-kilometer orbit. Now, DQ-1 must establish a robust retrieval algorithm

and conduct multi-source data validation on algorithmic outcomes, ensuring the system observation's precision.

   In this work, we studied the aerosol optical parameter retrieval algorithm of the spaceborne HSRL system. This algorithm has used the data of the attenuated backscatter coefficient of the perpendicular polarized channel, parallel polarized channel and molecular scattering channel of the DQ-1 ACDL system. We used the atmospheric temperature and pressure data in the Medium-Range Weather Forecasts Reanalysis v5 (ERA5) to calculate the molecular backscatter coefficient. To ensure the

accuracy of the retrieval results, the attenuated backscatter coefficient is first compared with the CALIPSO dataset. The retrieval results were further compared with the corresponding data products of CALIPSO, AERONET, and NASA MPLNET, where the errors and sources were analyzed. The comparison results confirm the accuracy of DQ-1 L2A datasets and retrieval algorithm. Lastly, We use data from DQ-1 to analyze the transport processes of Saharan dust and South Atlantic tropospheric





volcanic ash. This primary application area discloses the scientific significance of the high-performance system introduced by
the DQ-1 satellite in the context of global aerosol detection applications, rendering it an alternative approach to the CALIPSO
satellite.

## 2 Instrumentation and method

### 2.1 DQ-1 ACDL system

#### 2.1.1 System overview

The ACDL system includes a Integral Path Differential Absorption (IPDA) lidar system and a High Spectrum Resolution Lidar
(HSRL) system, capable of performing integrated satellite-based detection of atmospheric aerosols, clouds, and carbon dioxide
(Weibiao et al., 2023). The laser beam of the HSRL system has a wavelength of 532.245 nm, a pulse repetition frequency of
40 Hz, and the absorption line of iodine molecules corresponds to line 1110. As the laser is integrated with the differential
absorption system, the aerosol system operates with two distinct pulses, pulse A and pulse B, to observe the atmosphere
practically. Based on the energy, both of the pulses can be categorized and can be adjusted during the retrieval process. The
optical system has a Cassegrain-type telescope with a primary mirror of 1-meter diameter. The receiving system consists of
three optical channels: the perpendicular polarized channel, the parallel polarized channel, and the high spectral resolution
channel (Weibiao et al., 2023; Dai et al., 2023). The main parameters of the DQ-1 HSRL system are shown in Table 1. The
measured absorption spectrum lines of the iodine vapor filter on ACDL are shown in Figure 1. The principle of aerosol
scattering suppression by the iodine filter is also shown in Figure 1, based on the characteristic of aerosol Mie scattering spectra
having a narrow bandwidth compared to molecular Rayleigh scattering, to get rid of aerosol scattering efficiently. After the
signal is filtered, aerosol Mie scattering is absorbed, presenting a residual portion of molecular Rayleigh scattering. The aerosol
suppression ratio of more than 25 dB is obtained. Using a calibrated system, the detector output signals obtained at three
channels make up the L2A attenuated backscatter coefficients used in the retrieval algorithm.

#### 2.1.2 Retrieval algorithm

Prior to L2A data retrieval, some pre-processing steps are taken, including signal-to-noise ratio control, moving average, and
pulse averaging. To eliminate the heavy cloud-covered signal, is the primary focus of data quality control, and removal of
erroneous echoes under the surface and signals with poor signal-to-noise ratios. There is an energy difference between laser
pulses A and B, where the L2A data have been calibrated during the production. To improve the SNR of the raw data, pulses
A and B were interleaved horizontally prior to pulse averaging. To compare the design's horizontal resolution of 20 km, we
used 48 sets to configure the pulse averaging iterations, and the vertical resolution is 48 meters. The main retrieval processes
are presented through a flowchart, as shown in Figure 2. The ERA5 and DQ-1 L2A datasets are given, which serve as the
algorithm's initial point. The ERA5 temperature and pressure data corresponding to the DQ-1 satellite is incorporated into the





atmospheric model (Tenti et al., 1974) to compute molecular backscattering spectra. Subsequently, to determine the
convolution transmittance of the molecules, $T_m$, the spectra is convolved with iodine absorption spectra. We used the signal-
to-noise ratio to control the data quality. Subsequently, we applied signal smoothing to the pre-processed data, which is then
utilized in the following retrieval steps. The computation of the depolarization ratio employs data from both parallel and
perpendicular depolarized channels, while the backscatter coefficient is from three channels and $T_m$. The extinction coefficient
is computed from parallel and high spectral resolution channels, along with $T_m$. The specific mathematical equation is
presented as follows.

Based on the mentioned receiving system principles outlined in Section 2.1.1, the equation for the attenuated backscatter
coefficient is described as follows, with a perpendicular depolarized channel, parallel depolarized channel, and high spectral
resolution channel.

$$B^\perp(r) = [\beta_m^\perp(r) + \beta_a^\perp(r)] \times exp\left\{ -2 \int_0^r [\alpha_m(r) + \alpha_a(r)]\, dr \right\} \tag{2.1}$$

$$B_C^\parallel(r) = [\beta_m^\parallel(r) + \beta_a^\parallel(r)] \times exp\left\{ -2 \int_0^r [\alpha_m(r) + \alpha_a(r)]\, dr \right\} \tag{2.2}$$

$$B_H^\parallel(r) = [T_m(r)\beta_m^\parallel(r) + T_a(r)\beta_a^\parallel(r)] \times exp\left\{ -2 \int_0^r [\alpha_m(r) + \alpha_a(r)]\, dr \right\} \tag{2.3}$$

In Eqs. (2.1) ~ (2.3), the symbols $\perp$ and $/\!/$ represent signals characterized by perpendicular polarized and parallel polarized,
respectively. The ratio between them reflects the sphericity of the target. The subscripts $C$ and $H$ represent the signals of the
parallel channel and the high spectral resolution channel, with their ratio reflecting the proportion of aerosol Mie scattering
signals to the backscattering signals. $T_m(r)$ and $T_a(r)$ respectively represents the transmittance of the echo signal of molecular
Rayleigh scattering and aerosol Mie scattering after passing the iodine vapor filter.

Simultaneous Eqs. (2.1) ~ (2.3), calculate the aerosol backscatter coefficient as follows:

$$\beta_a(r) = \beta_m(r) \frac{[1 + \delta(r)]}{(1 + \delta_m)} \frac{[T_m(r) - T_a(r)]K(r)}{[1 - T_a(r)K(r)]} - \beta_m(r) \tag{2.4}$$

Where $\delta_m$ represents the depolarization ratio of molecules, and $\delta(r)$ represents the volume depolarization ratio, which is
dependent to the spherical state of the target. The volume depolarization ratio is expressed as:

$$\delta(r) = \frac{B^\perp(r)}{B_C^\parallel(r)} \tag{2.5}$$

Similarly, the atmospheric optical depth is defined as:

$$\tau(r_0) = \int_0^{r_0} (\alpha_a(r) + \alpha_m(r))\, dr = -\frac{1}{2} ln \left[ \frac{(1 - K(r_0)T_a(r_0))(1 + \delta_m)B_H^\parallel}{(T_m(r_0) - T_a(r_0))} \right] \tag{2.6}$$

Where $K(r_0)$ is the ratio of parallel channel to molecular channel, it can be expressed as:

$$K(r_0) = \frac{B_C^\parallel(r_0)}{B_H^\parallel(r_0)} \tag{2.7}$$





The optical depth of the atmosphere is defined as:

$$\tau(r_0) = \int_0^{r_0} \left(\alpha_a(r) + \alpha_m(r)\right) dr = -\frac{1}{2} ln \left[\frac{(1 - K(r_0)T_a(r_0))(1 + \delta_m)B_H^{\parallel}}{(T_m(r_0) - T_a(r_0))}\right] \qquad (2.8)$$

Differentiating the Eq. (2.8), the aerosol extinction coefficient can be expressed as:

$$\alpha_a(r_0) = \frac{\partial \tau(r_0)}{\partial r} - \alpha_m(r_0) \qquad (2.9)$$

The aerosol lidar ratio is expressed as:

$$S_a(r) = \frac{\alpha_a(r)}{\beta_a(r)} \qquad (2.10)$$

## 2.2 CALIPSO

The Cloud-Aerosol Lidar and Infrared Pathfinder Satellite Observation (CALIPSO) satellite, launched on April 28, 2006, is equipped with the Cloud-Aerosol Lidar with Orthogonal Polarization (CALIOP) instrument operating at a wavelength of 532

nm and 1064 nm. CALIOP continuously observes the Earth's atmosphere to monitor attenuated backscatter data with depolarization and color ratios. The CALIPSO Level 2 data includes vertical profiles of aerosol backscatter coefficient, extinction coefficient, and depolarization ratio. Various approved methodologies have been deployed to accurately monitor and observe the Earth's atmosphere (Mcpherson et al., 2010). The retrieval algorithm employed for Level 2 data processing has been refined frequently, with the updated version of V4.51. Because of the CALIPSO energy attenuation, the satellite is

currently malfunctioning, demanding the deployment of a new satellite platform to continue global observations of clouds and aerosols. Due to orbital decay resulting from the depletion of satellite propellant, this satellite was decommissioned on August 1, 2023.

## 2.3 AERONET

The AErosol RObotic NETwork (AERONET) is a ground-based aerosol remote sensing network established by the mutual

collaboration of NASA and LOA-PHOTONS (CNRS) (Holben et al., 1998). At the moment, this global automated observation network has coverage in major regions throughout the world. The network utilizes the automated sun photometers (SPAM) produced by CIMEL as the primary instrument to observe the atmosphere. The instruments at most sites collect data daily, which is then further processed by the instrumental setup. AERONET provides a valuable resource for multi-wavelength, continuous, and accurate aerosol optical thickness data products. These data products play a significant role in studying global

aerosol transport, aerosol radiative effects, validating radiative transfer models, and verifying satellite-based aerosol remote sensing results.


### 2.4 NASA MPLNET

The NASA Micro-Pulse Lidar Network (MPLNET) is a globally distributed network equipped with a polarized Micro-Pulse Lidar (MPL) system that has been operating continuously since 2000 (Pappalardo et al., 2014). With more than 70 well-established observational stations worldwide, MPLNET has several underlying surface conditions, allowing it to collect ongoing aerosol vertical profiles in different religions. The aerosol backscatter coefficient, extinction coefficient, and depolarization ratio can be measured by its system. Most of the MPLNET sites are strategically situated near the observation sites of the Aerosol Robotic Network (AERONET). The integration of MPLNET and AERONET is used to make a potential approach possible, and to facilitate the robust validation of satellite instruments and other scientific objects.

### 2.5 HYSPLIT model

The NOAA Hybrid Single Particle Lagrangian Integrated Trajectory (HYSPLIT) model is a widely used atmospheric transport and dispersion model developed by the National Oceanic and Atmospheric Administration (NOAA). It has been extensively employed for various applications, including air pollution studies, atmospheric research, emergency response planning, and radiological assessments. The HYSPLIT model employs a Lagrangian approach to trace aerosol movement and simulate the dispersion of pollutants or other atmospheric constituents over time. This model provides valuable insight into the long-range transport of pollutants, the dispersion patterns of hazardous substances, and the understanding of atmospheric dynamics on local, regional, and global scales (Crawford et al., 2016; Wang and Chen, 2014).

## 3 Validation of retrieval result

### 3.1 Validation with CALIPSO

We conducted a comparative validation between the CALIPSO and DQ-1 satellites on a closely aligned orbit on June 6, 2022. The ground tracks of both satellites are illustrated in Fig. 3. The two satellites follow orbital trajectories extending from the Central Asian region across the Arabian Peninsula to the open ocean south of Africa, with a spatial separation of 400 km and a temporal difference of 60 minutes. In order to verify the consistency of the raw data, Figures 4a and 4b describe the total attenuated backscatter coefficients of DQ-1 and CALIPSO. Within the geographical area of 40°N to 10°N, at an altitude below 5 km, desert dust aerosols were investigated in large proportion. The attenuated backscatter coefficient in this area falls within the range of $10^{-3}$ to $10^{-2}$ km$^{-1}$sr$^{-1}$. This geographical area intersects the Arabian Peninsula and the Iranian Plateau. At an altitude of 15 km, the distribution of stratocumulus clouds was observed, with the satellite's emitted laser failing to penetrate certain portions of the cloud cover. South of 10°N, as the two satellites entered the maritime region, there was no observed aerosol distribution. The cumulus clouds were distributed at an altitude of 3 km, and some mid-level clouds reached an altitude of 5 km. South of 20°N, at an altitude higher than 20 km, DQ-1 observed the distribution of stratospheric volcanic aerosols. Due to the impact of temporal and spatial inconsistencies, the results obtained about cloud detection from CALIPSO and DQ-1





have dissimilarities. Nevertheless, both the aerosol results exhibit similar trends, with numerical values ranging from $10^{-3}$ to $10^{-2}$ km$^{-1}$sr$^{-1}$. To further evaluate the differences of both the raw signals. Figure 4e presents the average raw signal with a latitude ranging from 20°N to 22°N, having molecular backscatter coefficient calculated from ERA5 temperature and pressure

data for analysis. The CALIPSO raw signal exhibits smaller values than DQ-1, primarily due to energy attenuation in the CALIPSO laser, with DQ-1's raw data better conforming to molecular backscattering signals above 4 km. The abrupt increase in the CALIPSO signal at a height of 9 km is due to the cloud's influence. To compare both the signal quality, high-altitude signals were employed to depict system noise, and the signal-to-noise ratio (SNR) of the total attenuated backscattered signals was analyzed, as illustrated in Figures 4c and 4d. The vertical resolution of the DQ-1 attenuated backscatter coefficient is 48m,

while CALIPSO exhibited a vertical resolution of 30 m below 8 km and 60 m between 8 km and 20 km. While the value of CALIPSO's aerosol signal SNR varied from 10 to 15 dB, the SNR of DQ-1's aerosol signal exceeded 15 dB. Additionally, DQ-1 has maintained a high-altitude molecular scattering SNR above 5 dB, whereas CALIPSO's high-altitude molecular scattering signal SNR has a value of less than 5 dB. In conclusion, both satellites give consistent raw data results in close orbits. However, due to laser energy attenuation, CALIPSO has a weaker echo signal than DQ-1. DQ-1 operating at a higher resolution,

achieves a better signal-to-noise ratio, with a molecular echo signal more closely aligned with ERA5 calculations.

Figure 5 shows a comparative analysis of the retrieval results obtained from both systems. Figures 5a and 5b show the latitudinal distribution of aerosol extinction coefficients with a $10^{-4}$ m$^{-1}$ value. Figures 5c and 5d illustrate the latitudinal distribution of backscatter coefficients for both systems, with a value of $10^{-5}$ m$^{-1}$sr$^{-1}$. Both systems vary in terms of clouds result, whereas CALIPSO's weak energy output cannot obtain high-altitude cloud retrieval results. Furthermore, at a height

below 2 km in the lower troposphere, at a spanning of 5°N to 15°N, CALIPSO has obtained significantly higher retrieval values, discriminating the aerosol types. In Figures 5i and 5j, we chose two satellites for the profile comparison of the aerosol extinction coefficient and backscatter coefficient at a latitude of 22°N. The backscatter coefficient was retrieved from both satellites with a value of $10^{-6}$ m$^{-1}$sr$^{-1}$ at a height lower than 4 km, whereas the aerosol extinction coefficient was retrieved with a $10^{-4}$ m$^{-1}$ value. This indicates a relatively dense distribution of aerosols. Besides atmospheric variations caused by specific

spatiotemporal effects, both variable's trends and numerical distributions are closely related. Figures 5e and 5f depicts the particulate depolarization ratio. The value of depolarization ratio at low altitude obtained from DQ-1 is 0.2, demonstrating the nature of mixed dust. Similarly, the depolarization ratio of clouds at a high altitude of 15 km measures a value of 0.4, presents the characteristics of cirrus clouds composed of ice crystals. The retrieval results of the aerosol depolarization ratio from CALIPSO exhibit a value of more than 0.3, with some higher values of 0.5. It is worth noting that these values are observed

particularly above the airspace in a desert-covered region of Iran and the Arabian Peninsula. According to the literature review, the value of depolarization ratio higher than 0.4 is typically associated with ice crystal particles (Burton et al., 2012b; Sakai et al., 2003; Wiegner et al., 2011). In contrast to the distinct features observed in the low-altitude atmosphere of the Central Asian desert in June, the observed discrepancy is attributed to errors resulting from the depolarization ratio retrieved by the laser energy attenuation in CALIPSO. Figures 5g and 5h depict the latitudinal distribution of aerosol lidar ratios. Both DQ-1 and

CALIPSO indicate that the lidar ratio of aerosol particles is 50 sr$^{-1}$, describing the characteristics of mixed dust aerosols. Due





to the proximity of the CALIPSO orbit to oceanic regions, below 3 km altitude and at a latitude below 18°N, CALIPSO retrieval results in sea salt aerosols, with numerical values representing a lidar ratio of 20 sr$^{-1}$ and a depolarization ratio less than 0.1. Considering the spatiotemporal and laser energy differences between the two satellites, the retrieval results of DQ-1 and CALIPSO are consistent. DQ-1, operating at high-resolution conditions, achieves a high signal-to-noise ratio and presents

reliable retrieval results.

## 3.2 Validation with MPLNET

The results obtained from ground-based observation platforms exhibit accuracy and comprehensive temporal coverage. To ensure the accuracy of aerosol optical properties retrieval using DQ-1, we compared aerosol optical parameter data products from NASA MPLNET ground-based stations. This comparison has three surface types: land, ocean, and coastal regions.

Profile-averaged findings inside a circle with a radius of 100 kilometers, centered on MPLNET stations, were explicitly chosen for DQ-1 data. Specifically, choosing DQ-1 data, profile-averaged results in a region of 100 kilometers radius centered on MPLNET stations were selected. For MPLNET data, we utilized the average MPLNET profile within 15 minutes of DQ-1 transit. Figure 6 illustrates the validation results. The change in lower atmospheric aerosols is fast and characterized by high local heterogeneity, whereas the signal-to-noise ratio at a high altitude is comparatively low. Hence, we selected the data at an

altitude range of 1 km to 8 km for comparison. Three selected distinct surface types, namely, land, coastal, and oceanic regions, correspond to the Apalachin, El Arenosillo, and Santa Cruz Tenerife sites. The location of these three sites and the satellite trajectory are illustrated in Figures 6c, 6f, and 6i. The difference between the satellite and ground-based retrieval results is quantified, as shown in Figure 7.

The Apalachin site is inland in the United States, where satellite passes during daytime hours. At this site, aerosol backscatter

coefficients obtained from satellites and ground-based observations exhibit a relatively good correspondence at an altitude range of 3 km to 5 km. The relative discrepancy between the two observations is less than 25% within this range. However, below 3 km, the measurements are influenced by clouds, increasing the error between the two datasets. Both satellite and ground-based observations indicate a decreasing trend in the backscatter coefficient with an altitude of more than 5 km. Above 5 km, ground-based data exhibit rapid variations, leading to increased variations between the two datasets. The El Arenosillo

site is situated along the southwestern coast of Spain, and the vertical profile comparison results are depicted in Figures 6d and 6e. Both observational methods indicate that the aerosol distribution exhibits relatively low variation within the altitude of 1 km to 6 km. Above 6 km, there is a decline in aerosol concentration with increasing altitude. The distribution of aerosol backscatter and extinction coefficients is consistent, with the maximum values occurring at an altitude of 3.5 km. The relative error between the two results is less than 25% at an altitude of more than 2.5 km. The Santa Cruz Tenerife station is situated

in the Canary Islands, northwest of Africa, near the ocean. Except for the altitude range between 3 km and 5 km, the profiles of aerosol optical parameters from both satellite and ground-based results remain relatively consistent. The aerosol extinction and backscatter coefficient profiles from both sources exhibit a high degree of agreement when accounting for spatial disparities. The relative errors between the two results are less than 50% for the aerosol extinction coefficient and less than





25% for the backscatter coefficient. Comparative analysis of NASA MPL ground-based data products, when there is no
influence of cloud, reveals that the relative errors in aerosol extinction and backscatter coefficients between the two sources
are about 25%. This further validates the accuracy of the DQ-1 satellite retrieval algorithm.

### 3.3 Validation with AERONET

Figure 8 illustrates a scatterplot comparison between aerosol optical depth data obtained from DQ-1 and AERONET Level 2.0
from June to December 2022. The DQ-1 data represents the average aerosol optical depth within a circular region of a radius
of 100 km, centered on AERONET sites, derived through the retrieval process. The data from AERONET stations represent
ground-based measurements obtained within a 15-minute time window when DQ-1 passes over these sites. The difference in
elevation between the sub-satellite point and the AERONET station can introduce errors (Omar et al., 2013). To mitigate such
errors, we excluded the deviated results at more than 200 meters from the station's altitude on the ground. To exclude cloud
signals, we subject the DQ-1 aerosol optical depth data to cloud signal removal using the backscatter ratio for better quality
enhancement. When the backscatter ratio exceeds 10, we consider it to indicate cloud signal, which is excluded from
calculating aerosol optical depth (Ke et al., 2022). Aerosol optical depth measurements at a wavelength of 532 nm were missing
at some AERONET stations. Hence, we employed data at a wavelength range of 500 nm to 550 nm as a substitute.
The results obtained from a first-order polynomial regression analysis of the scatterplot data revealed a variance of 0.803 and
a root mean square error (RMSE) of 0.1231. Considering the spatiotemporal differences between the two datasets, the satellite-
based and ground-based observations exhibited a commendable level of agreement. There are specific data points deviating
from the fitted regression line. These deviating data points share a common characteristic: satellite-based data tends to exhibit
higher values. This difference can be attributed to the incomplete removal of cloud signals during retrieval. Such issues have
also been highlighted in prior studies (Omar et al., 2013). Due to the current status of DQ-1's Level 2A data processing,
completed only from June to December 2022, and ongoing data processing efforts for specific AERONET datasets, the
available quantity of matching results is limited. Further validation through satellite-ground comparisons over an extended
period can be pursued.

### 4 Preliminary application of retrieval result

After validation with various well-established observational platforms, our research has provided initial verification of the
observational accuracy of the DQ-1 HSRL system. The following work will showcase some of the scientific applications that
can be achieved with DQ-1, unveiling the immense potential and scientific value of the high-performance system carried by
this satellite in comprehensive atmospheric environmental monitoring.



## 4.1 Observation of aerosol transport in east Africa

In July 2022, DQ-1 observed the propagation of Saharan dust from the northeastern Sahara region of Africa towards the Atlantic region. The optical parameters along the aerosol transport path from 3rd July to 8th July were retrieved. Additionally,
a six-day backward trajectory analysis using the NOAA HYSPLIT Model was conducted on July 8th over the northeastern Atlantic off the coast of South America. Within this six-day duration, the dust covers a distance of 4000 km, with an average speed of 30 km/h. These results are presented in Figure 9. From July 3rd to July 4th, DQ-1 observed the aerosol layer at an altitude of 8 km, with the backscatter coefficient ranging from $10^{-5}$ to $10^{-4}$ $m^{-1}sr^{-1}$. Following a westward transport over two days, July 7th to July 8th, the altitude distribution is reduced to 3 km, and the backscatter coefficient has fallen below $10^{-5}$ $m^{-1}sr^{-1}$. During this transport process, there was an observable trend of aerosol settling over the ocean, reducing the altitude
distribution range and backscatter coefficient. Figures 9b and 9c show the lidar and volume depolarization ratios, respectively. On July 3rd, measurements over West Africa indicated a lidar ratio distribution centered around 50 $sr^{-1}$ and depolarization ratio values within the range of 0.3 to 0.4, representing mixed dust (Burton et al., 2012a; Groß et al., 2011; Groß et al., 2013). During the transport process, the value of the lidar ratio was constant at 50 $sr^{-1}$, while the volume depolarization ratio was
reduced from 0.3 to 0.15. This trend suggests that dust gradually mixes with other aerosol types as the aerosols were transported. On July 8th measurements, the lidar ratio from the surface to 2 km altitude exhibited values centered around 20, describing the characteristic of sea salt (Burton et al., 2012a; Groß et al., 2011; Groß et al., 2013). Above 2 km, the lidar ratio is about 50 $sr^{-1}$, indicating a consistent presence of mixed dust. These observations have revealed a stratified distribution of aerosols in this region. The satellite-based retrieval results have presented the spatial variations in dust's position and optical properties during
their transport. DQ-1 observed that as these aerosols were transported, their altitude, backscatter coefficient, and depolarization ratio decreased at the lidar ratio's constant value.

## 4.2 Observation of stratospheric aerosol distribution over the south Atlantic

From January 14th to 15th, 2022, the Tonga volcano experienced two significant eruptions, releasing a substantial volume of volcanic ash, gases, and water vapor into the upper atmosphere, forming extensive cloud formations. The volcanic ash reached
an altitude of 20 km (Yufeng et al., 2022). DQ-1 observations over the South Atlantic have revealed various effects and distribution of volcanic ash in the stratosphere. As DQ-1 data is available from June to December 2022, we substituted data from February to June 2022 with CALIPSO data. It should be noted that DQ-1 and CALIPSO have differences in laser energy. We corrected the data presented in the figures for energy variations. Figure 10 presents the observed attenuated backscatter coefficient from January to December 2022 within the stratosphere over the South Atlantic Ocean, using both CALIPSO and
DQ-1. The area in the figure, ranging from 20° S to 30° S, falls within CALIPSO's South Atlantic Anomaly (SAA) region. The laser energy is weaker in this region, resulting in a low signal-to-noise ratio. On January 1st, CALIPSO detected no aerosol distribution within the stratosphere. Following a volcanic eruption in mid-January, on February 1st, significant signals emerged at an altitude of 20 km, with attenuated backscatter coefficients reaching $10^{-3}$ $km^{-1}sr^{-1}$. From February to June, stratospheric





aerosols' distribution gradually extended from 5° S to 20° S latitude, with the attenuated backscattering reducing constantly to
less than $10^{-4}$ km$^{-1}$sr$^{-1}$. By May 1st, due to insufficient laser energy, CALIPSO received weak volcanic backscatter signals,
making them difficult to distinguish from system noise. By June 1st, DQ-1 initiated observations in this region, revealing the
presence of stratospheric aerosols at an altitude of more than 20 km, spanning from 0° S to 40° S latitude. Due to the diffusion
of volcanic aerosols in the stratosphere and the advantages in laser energy of the DQ-1 system, the results from DQ-1 indicate
a broader distribution range of volcanic ash. By August 1st, volcanic ash had extended southward to 50° S latitude, with an
altitude of less than 20 km. From September to December, aerosols are consistently reduced and dissipated within 30° S to 60°
S latitude. By December 1st, their distribution spanned from 30° S to 50° S latitude, decreasing altitudes to around 15 km.

**5 Conclusion**

This research has studied satellite-based retrieval algorithms and multi-source data validation of algorithmic results to obtain
satellite-based aerosol optical properties accurately. The attenuated backscatter coefficient obtained from DQ-1 has been
validated against the product of CALIPSO and molecular backscatter coefficients. The results indicated that DQ-1 exhibited a
higher signal-to-noise ratio and conforms to the results of trends in molecular scattering. The aerosol optical parameters
retrieved from DQ-1 have been validated against those from CALIPSO. Taking spatiotemporal differences into account, both
payload's retrieval results exhibit consistency, with DQ-1 yielding more reliable depolarization ratio outcomes. Data products
from MPLNET stations, representing the three underlying surface types, have been selected for satellite-to-ground validation
purposes using DQ-1. Both datasets have yielded consistent trends in the extinction and backscatter coefficient profiles, with
a relative error of 25% after excluding cloud interference. Comparing the AOD from DQ-1 with the AOD obtained from
AERONET within the selected spatiotemporal domain, a correlation analysis has yielded a value of $R^2$ 0.803 and RMSE of
0.1231, indicating a strong correlation between the two datasets. The validation process, conducted in conjunction with
CALIPSO, MPLNET, and AERONET, has ensured the accuracy of the raw data and retrieval results obtained by the system.
This paper has undertaken an initial application into the transport of dust aerosols in East Africa on the retrieval-based results.
The research findings indicate that the lidar ratio remains constant during the aerosol transport process, while the depolarization
ratio and backscatter coefficient are decreased. The study employed the attenuated backscatter coefficient from CALIPSO and
DQ-1 to investigate the South Atlantic stratospheric volcanic ash in 2022. The research findings have unveiled the latitude and
altitude variations in the vertical distribution of volcanic ash. The mentioned preliminary scientific applications demonstrate
that the DQ-1 spaceborne HSRL system is capable of accurate global observations. The system can be used to obtain far more
efficient retrieval techniques and comprehensive multi-source data validation with further scientific applications. Therefore, it
can be a suitable alternative for satellite-based systems like CALIPSO. The above preliminary scientific applications
demonstrate that the HSRL system is capable of accurate global observations.

**Data availability**

The DQ-1 L2A data was not publicly available at the time our manuscript was submitted. We were able to access the data by joining as part of the scientific team for DQ-1. The ERA5 dataset is downloaded via the website https://cds.climate.copernicus.eu/ (last access: 10 October 2023). The AERONET dataset is downloaded via the website https://aeronet.gsfc.nasa.gov/ (last access: 10 October 2023). The MPLNET dataset is downloaded via the website https://mplnet.gsfc.nasa.gov/ (last access: 10 October 2023). The CALIPSO dataset is downloaded via the website

https://asdc.larc.nasa.gov/project/CALIPSO/ (last access: 10 October 2023).

**Author contributions**

C. Zha contributed to algorithm development and data analysis and wrote the manuscript. L. Bu, Q. Wang contributed to scientific discussions and reviewed the manuscript. A. Mubarak and P. Liyanage contributed to modifying the grammar of the manuscript. All the co-authors reviewed and edited the manuscript.

**Competing interests**

The authors declare that they have no conflict of interest.

**Acknowledgments**

We thank the Shanghai Institute of Optics and Fine Mechanics, Chinese Academy of Sciences for providing the Level 2A data of DQ-1. We thank the AERONET Principal Investigators and their staff for establishing and maintaining the AERONET at

the 40 sites used in this work. We thank NASA Langley Research Center for providing the data sets of CALIPSO. We are also thankful to the MPLNET PIs for their effort in establishing and maintaining the Apalachin, El Arenosillo, and Santa Cruz Tenerife sites. We also thank the European Centre for Medium-Range Weather Forecasts (ECMWF) for providing the ERA5 dataset.

This research was funded by the National Natural Science Foundation of China (Grant No. 42175145) and the Shanghai

Aerospace Science and Technology Innovation Foundation (SAST2022-039).

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





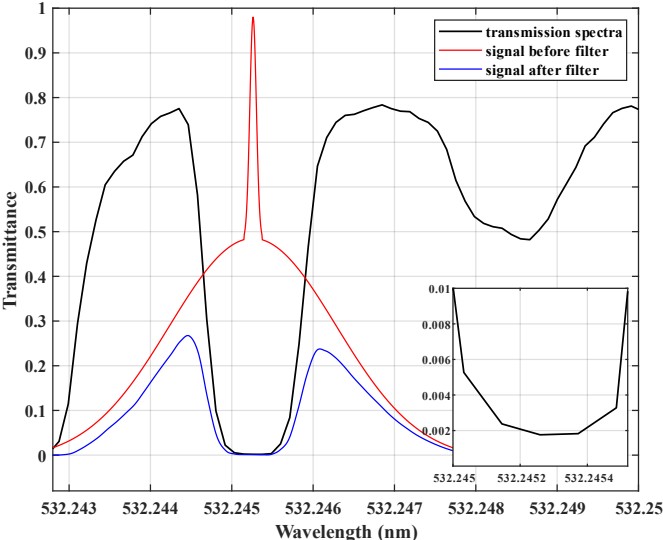

**Figure 1 The actual measured transmittance spectra of the on-board iodine vapor filter of the DQ-1 satellite. The subfigure in the lower right corner displays the transmittance spectrum in 1110 line. The red solid line delineates the spectral of the echo signal prior to filter, the blue solid line delineates the spectral of the echo signal after the fliter.**



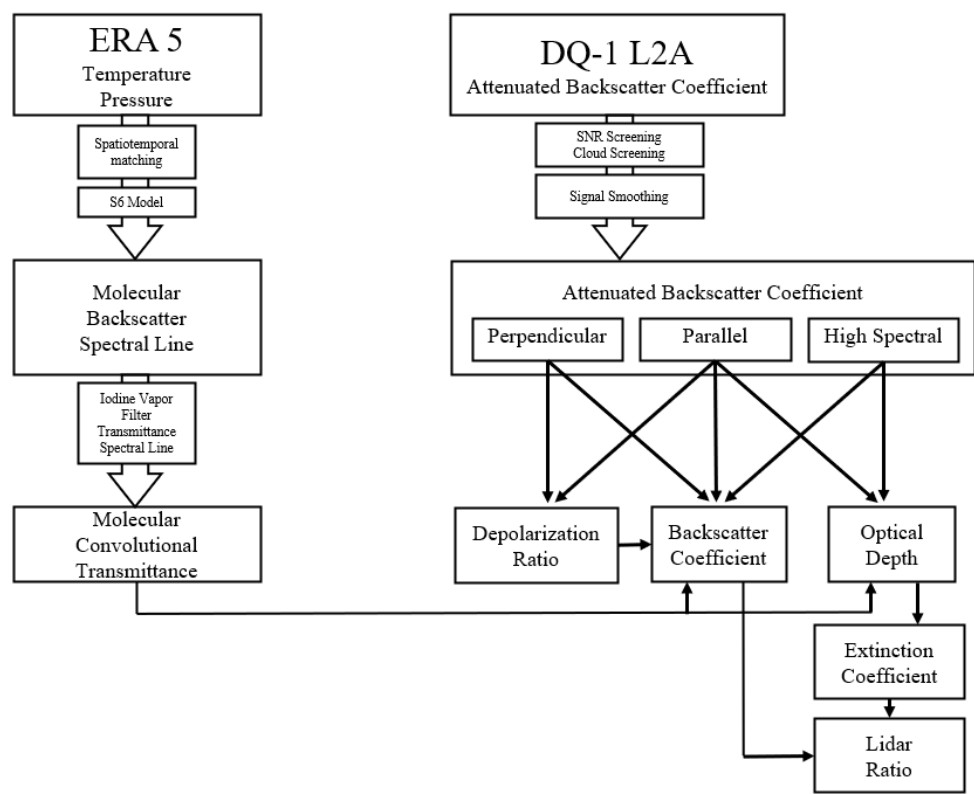


**Figure 2 Flowchart of the DQ-1 retrieval algorithm**



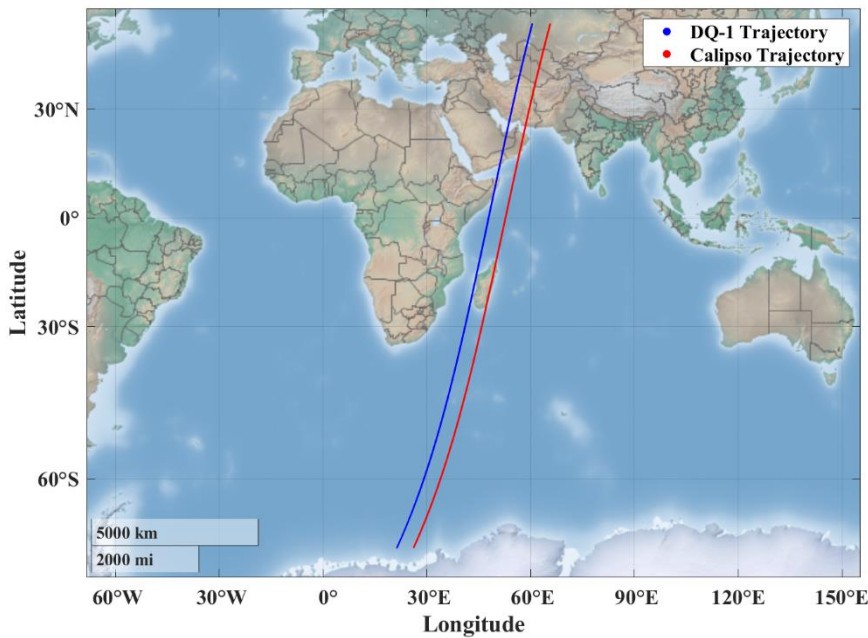

**Figure 3 The selected satellite trajectory of DQ-1 and CALIPSO in June 6, 2022. The DQ-1 trajectory is indicated by the blue solid line, while the CALIPSO trajectory is represented by the red solid line. Time difference between the two trajectories is 60 minutes, spatial separation amounts to 400 kilometers.**



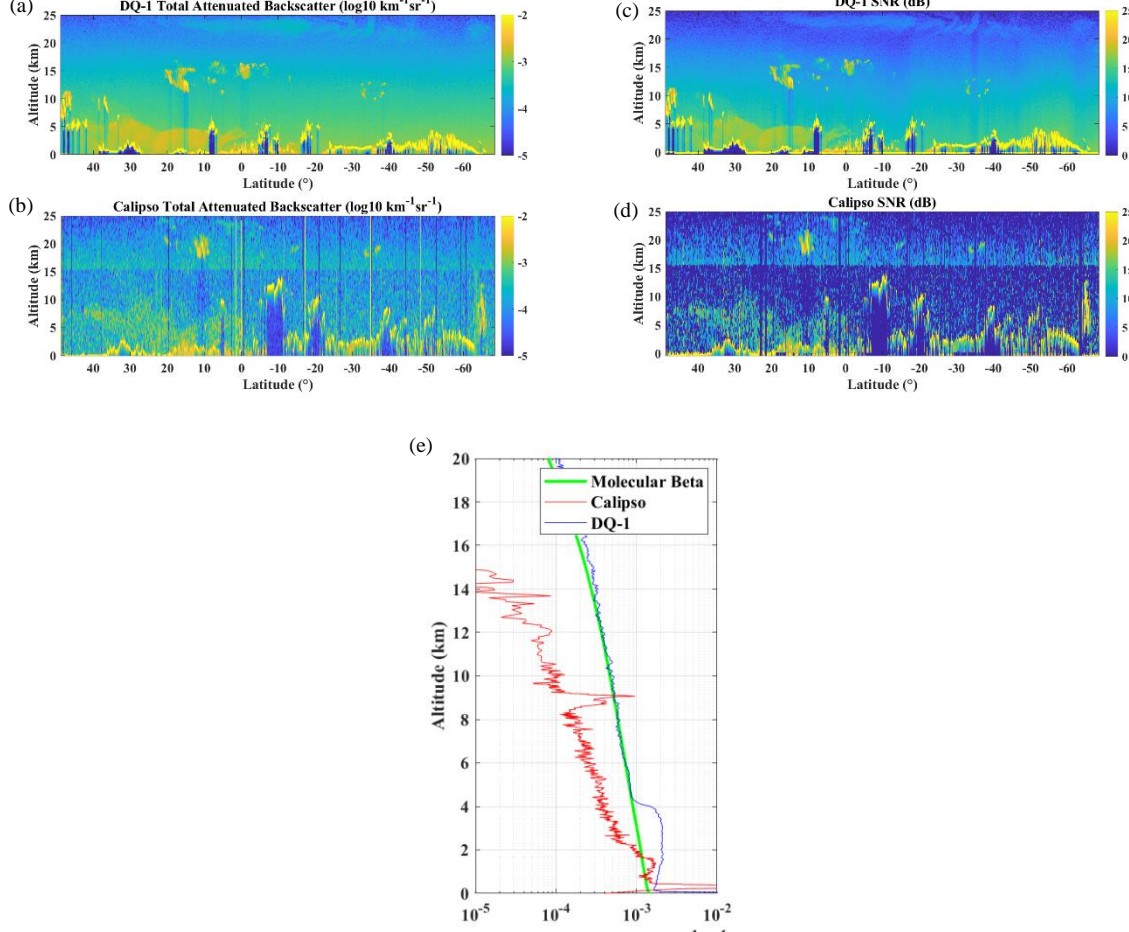


**Figure 4 Profile comparison of DQ-1 and CALIPSO on June 6, 2022: (a) Total attenuated backscatter coefficient profiles obtained by the DQ-1; (b) Total attenuated backscatter coefficient profiles obtained by the CALIPSO; (c) SNR of DQ-1 total attenuated backscatter; (d) SNR of CALIPSO total attenuated backscatter; (e) Comparison of total attenuated backscatter single profile, the blue solid line represents the DQ-1 results, the red solid line represents the CALIPSO results, and the green solid line depicts the computed molecular backscatter coefficient.**








**Figure 5 Profile comparison of DQ-1 and CALIPSO on June 6, 2022: (a) Extinction coefficient profiles of aerosols obtained by DQ-1; (b) Extinction coefficient profiles of aerosols obtained by CALIPSO; (c) Backscatter coefficient profiles of aerosols obtained by DQ-1; (d) Backscatter coefficient profiles of aerosols obtained by CALIPSO; (e) Particulate depolarization profile obtained by DQ-1; (f) Particulate depolarization profile obtained by CALIPSO; (g) Lidar ratio profile obtained by DQ-1; (h) Lidar ratio profile obtained by CALIPSO; (i) Comparison of aerosol backscatter coefficient single profile; (j) Comparison of aerosol extinction**



**coefficient single profile. The blue solid line represents the DQ-1 results, the red solid line represents the CALIPSO results, and the green solid line depicts the computed molecular backscatter coefficient, the shaded area represents the standard deviation of different detection heights**





**Figure 6 Comparison of single profiles between DQ-1 and NASA MPLNET, The red solid line represents MPLNET results, blue solid line represents DQ-1 results, the shaded area represents the standard deviation of different altitudes. (a) Comparison of aerosol extinction coefficient profiles between DQ-1 and MPLNET Apalachin site on August 16, 2022; (b) Comparison of aerosol backscatter coefficient profiles between DQ-1 and MPLNET Apalachin site on August 16, 2022; (c) Trajectory of DQ-1 orbit and MPLNET Apalachin site location. (d) Comparison of aerosol extinction coefficient profiles between DQ-1 and MPLNET El_Arenosillo site on**



**Jun 13, 2022; (e) Comparison of aerosol backscatter coefficient profiles between DQ-1 and MPLNET El_Arenosillo site on Jun 13, 2022; (f) Trajectory of DQ-1 orbit and MPLNET El_Arenosillo site location; (g) Comparison of aerosol extinction coefficient profiles between DQ-1 and MPLNET Santa_Cruz_Tenerife site on Aug 22, 2022; (h) Comparison of aerosol backscatter coefficient profiles between DQ-1 and MPLNET Santa_Cruz_Tenerife site on Aug 22, 2022; (i) Trajectory of DQ-1 orbit and MPLNET Santa_Cruz_Tenerife site location.**





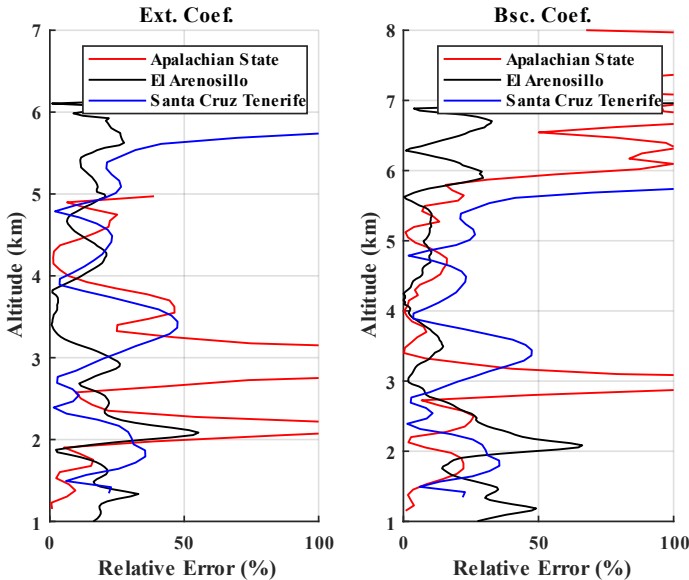


**Figure 7 The relative errors between the three MPL sites and DQ-1. The red line represents the relative error between the profiles of Apalachian State station and the DQ-1, the black line represents the relative error between the profiles of El Arenosillo station and the DQ-1, the red line represents the relative error between the profiles of Santa Cruz Tenerife station and the DQ-1.**






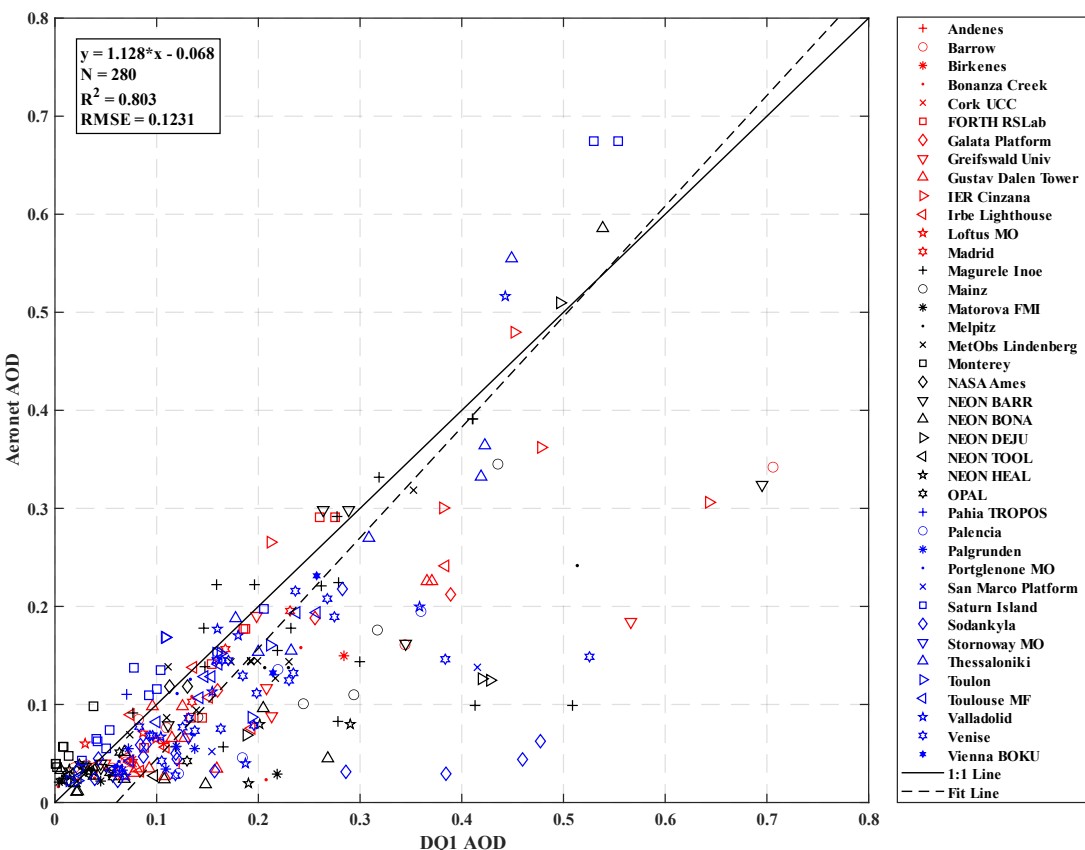

**Figure 8 Scatter plot of aerosol optical depth measured by DQ-1 and AERONET. Distinct shapes and colours are employed to indicate the scatter of AERONET and DQ-1 aerosol optical depth from various measurement sites. The solid black line represents the unity line (1:1), the black dashed line represents the outcome of a linear regression between the two datasets. The coefficient of**
**determination (R-squared) is computed as 0.803, based on 280 fitted data points.**





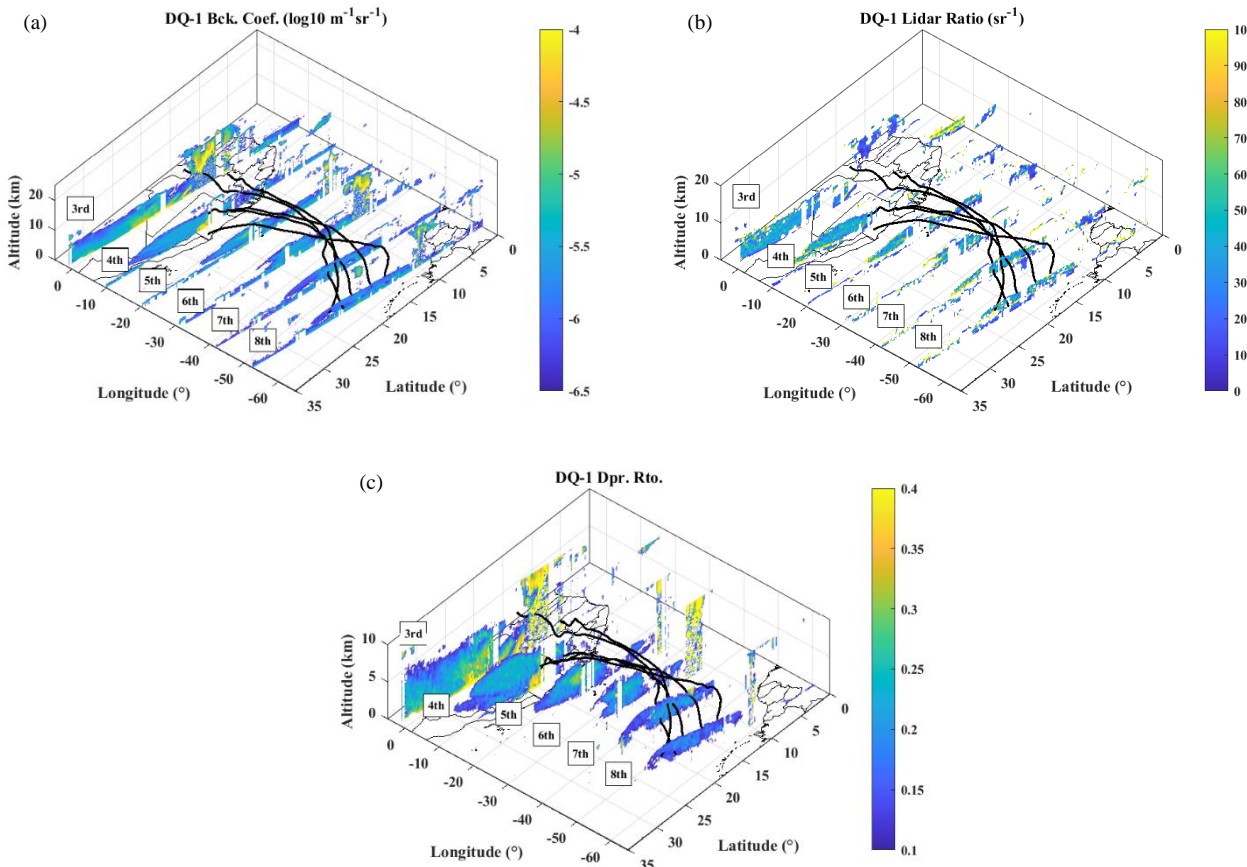

**Figure 9 Aerosol optical parameters retrieval results from DQ-1 over the Atlantic Ocean and corresponding HYSPLIT backward trajectory analysis. The figure presents observations from six orbits spanning July 3rd to July 8th, with dates indicated on the left side. The black solid lines denote the aerosol transport paths derived from the HYSPLIT backward trajectory analysis. (a) DQ-1 aerosol backscatter coefficient; (b) DQ-1 lidar ratio; (c) DQ-1 depolarization ratio;**



**Figure 10 Observed volcanic aerosol attenuated backscatter profile in the stratosphere over the South Atlantic in 2022. The left axis**
**displays date, while the bottom axis displays latitude, with an altitude range of 18 to 26 km, the results from January 1st to May 1st**
**are derived from CALIPSO, while the results from June 1st to December 1st are derived from DQ-1.**



| Laser Wavelength | 532.245nm |
|---|---|
| Laser Energy | ≥150mj |
| Laser Frequency Stability | 1MHz@10000s |
| Laser repetition frequency | 40 Hz |
| Telescope aperture | 1000mm |
| HSRL filters | iodine vapor filter, 1110 line<br>aerosol signal suppression ratio ≥ 25 dB |
| Receiving channel | parallel polarized channel<br>perpendicular polarized channel<br>high spectral resolution channel |
| Measurement accuracy | 15% uncertainty with 20km parallel resolution |

**Table 1 Main parameters of the DQ-1 HSRL system.**