# Peer review of "Aerosol Optical Properties Measurement using the Orbiting High Spectral Resolution Lidar onboard DQ-1 Satellite: Retrieval and Validation"

_Atmospheric Measurement Techniques, 2023_

## Referee Comment (RC2)

**General comments**

This paper reports important results from the high spectral resolution lidar flying on the DQ-1 satellite. These are the first HSRL aerosol retrievals from space, so the paper is quite significant. The paper presents initial retrieval results and comparisons with other observations but requires additional details and explanation before it is publishable.

To better understand the retrieval algorithm and the nature of the data products, more detailed description of the instrument and the signal processing steps applied are necessary. Several figures require more explanation. Since the paper has a focus on validation, methods use to calibrate the instrument and calibration accuracy should be explained, as well as the expected retrieval accuracy. Expressing ratios in dB is confusing. While common in the radar community, in the lidar community ratios are nearly always expressed as linear ratios rather than in dB. I strongly recommend that linear ratios be used rather than dB.

When validating instrument retrievals, one would like to compare with data which is more accurate, or equally accurate. This is a problem for DQ-1 because there may not be any suitable profile data available for comparisons. Discussion of comparisons with MPLnet and CALIPSO in Section 3 should acknowledge that aerosol extinction retrievals from the MPLnet and CALIPSO elastic backscatter lidars have significant uncertainties, which may be larger than the uncertainties of the DQ-1 retrievals. Thus the discrepancies between MPLnet and DQ1 could be entirely due to uncertainties in the MPLnet retrieval and it could be difficult to say what the accuracy of the DQ1 retrieval is. Comparisons of column AOD from AERONET, on the other hand, represent a true validation as AERONET AOD is quite accurate.

**Detailed comments**

Lines 49-51: What instrument was used to conduct the "observational experiments"? Was this the "airborne scaling system for ACDL" mentioned later? Please clarify. Does "multi-source data" refer to ground-based instruments? What type of instruments provided data?

Line 65: CALIPSO was "retired" in fall 2023 but not because of fuel consumption. CALIPSO science operations were terminated in August 2023 because the satellite's solar arrays could no longer generate enough electrical power to operate the CALIOP lidar.

Lines 72-74: Plans for AOS no longer include an HSRL instrument. AOS information on the vertical motion of clouds will come from Doppler radar.

Line 79: EarthCARE is anticipated to be launched in spring 2024.

Line 85: Is the "airborne scaling system for ACDL" an airborne simulator of the DQ-1 HSRL?

Line 95: "to ensure the accuracy" – HSRL retrievals should be more accurate than elastic backscatter lidars such as CALIOP and MPLNET. Intercomparisons are useful but elastic

backscatter lidars have significant retrieval errors and are not suited to validate HSRL accuracy. The comparison of DQ-1 AOD with Aeronet is more helpful.

Section 2. Instrumentation and Method

Very little has been published in English on this instrument. Additional description is necessary to understand the nature of the signals to which these retrievals are applied.

Line 118: Is the suppression of the aerosol signal only 25 dB? Using an iodine filter, the suppression of the aerosol signal can be much greater than 25 dB. Can the authors discuss?

Lines 121-122: Some discussion of the pre-processing steps is required. What is meant by "signal to noise ratio control" and what type of moving average and pulse averaging is applied?

Table 1 gives laser energy as "> 150 mJ" but the manuscript says laser pulses A and B have different energies. What is the energy of each pulse, A and B, and how they are averaged together? What is the time delay between these two 532 nm pulses? What does it mean they can be "categorized and adjusted during the retrieval process" (Line 110) The operations described in lines 125-127 are not clear. What is meant by "48 sets".

How is "measurement accuracy" in Table 1 defined? Is this the random error of the parallel channel signal or does it include calibration errors? Measurement accuracy depends on many factors, including background lighting, altitude, and averaging. Under what conditions is the measurement accuracy of 15% achieved? Is this before or after noise reduction is applied?

Important parameters are missing from Table 1. The manuscript mentions "energy differences" between CALIOP and DQ-1 several times but does not explain what the difference is. Lidar sensitivity depends on more than laser pulse energy. To gain more understanding of data quality and to better interpret intercomparisons, parameters such as receiver field of view, laser linewidth, bandwidth of the Fabry-Perot etalon, detector type (PMT, APD?, other?), detection scheme (analog or photon counting), and dynamic range of the detection system should be given. What is the view angle of the lidar - pointed at nadir or off-nadir?

Line 124 says "L2A data have been calibrated during production" How is calibration of the three signals accomplished? What is the accuracy of this calibration? What is the accuracy of the ratio of the parallel channel to molecular channel and of the volume depolarization ratio.

Line 127: The DQ-1 L2A dataset is an input to the algorithm. Explain what the L2A dataset is and what processing steps are used to produce the L2A.

Line 131: Explain how SNR is used to control data quality. Explain what kind of signal smoothing was applied to the pre-processed data. How is this different from the averaging applied as a pre-processing step?

The authors should discuss the magnitude of $T_m$ and $T_a$ in equation 2.3 and refer to Fig 1. From Fig 1 it looks like $T_a$ is about 0.002 and $T_m$ is perhaps 40-50%. Since this is a validation paper,

the authors should address how $T_m$ and $T_a$ vary on-orbit and whether this variation is a source of retrieval uncertainty.

Line 159: Is the extinction profile really computed from a simple derivative as described in Eq 2.9? This method is extremely sensitive to signal noise.

Line 169-171: These two sentences are not correct. In the last few years, the CALIOP laser was producing an increasing number of laser pulses with near zero energy. As explained above, science operations were terminated in August 2023 because the orbit had precessed to the east and the satellite's solar arrays could no longer generate enough electrical power to operate the lidar.

Section 3. Validation

Additional detail is needed on how CALIPSO data is used in the intercomparison with DQ-1. "Energy attenuation of the CALIPSO laser" is mentioned a number of times and pointed to as the source of various discrepancies. It is not clear what is meant by "laser energy attenuation". This term needs to be explained.

Line 207: There are no stratocumulus clouds at 15 km altitude. There are clouds at 15 km at about 15N which attenuate the lidar signal and look like dense cirrus.

Line 218: The method used to estimate SNR needs to be described in more detail. Variability of the signal due to noise must be separated from variability of the atmosphere.

Line 221: What is the "aerosol signal SNR" and how is it computed? For a backscatter lidar, the SNR of the component of the return signal due only to aerosol scattering doesn't make much sense.

Line 243-244: Please explain why the difference in depolarization is attributed to CALIPSO and not to DQ-1? What is meant by "the depolarization ratio retrieved by the laser energy attenuation in CALIPSO."

Line 245: CALIPSO lidar ratios are estimated, not retrieved, so they do not themselves provide validation of lidar ratios retrieved from DQ-1.

Line 253: Comparisons with MPLnet set a bound on the accuracy of DQ-1 retrievals but do not "ensure the accuracy of aerosol optical property retrieval using DQ-1" because DQ-1 retrievals should be more accurate than those from MPLnet.

Line 333: Please describe how the data was 'corrected for energy variations'. If the attenuated backscatter profiles are properly calibrated, further correction for energy variations should not be necessary.

Line 353 states that DQ-1 yields more reliable depolarization ratios than CALIOP. This statement needs more discussion and needs to be supported by analysis. How was it established that DQ-1 is "more reliable", does this mean more accurate?

**Comments on figures**

There is a problem in the way the CALIOP profile has been plotted in Fig 4e. Does "raw signal" refer to the Level 1 attenuated backscatter profiles? Inspection of CALIOP browse images shows the mean 532 nm attenuated backscatter at 20 km is roughly 1E-4 /km/sr whereas Fig 4e indicates the attenuated backscatter is already below 1E-4 /km/sr at 10 km altitude. Please explain how the CALIOP profile in Fig 4e was computed.

Figure 8 shows a significant high bias in many of the AOD retrievals from DQ1. The authors attribute this high bias to cloud contamination. Have the authors demonstrated this? Improved removal of clouds should improve the agreement or might reveal other sources of bias.

In Section 4.1 it is stated that the backscatter coefficient and volume depolarization of Sahara dust decreased during transport across the Atlantic, while the dust lidar ratio remained constant. This is difficult to tell from Figure 9. It would be helpful to add additional plots which show these trends (or lack of trend) more clearly.

What do the data curtains in Figure 10 represent? Is each curtain a single orbit (at what longitude?) on the first day of the month, or the average of several orbits (over what range of longitudes) averaged over a month of data?

**Minor issues and technical corrections**

Line 56: "This global information values …" should be "This global information is valuable …"

Line 64: Does Chiang et al. 2011 describe a retrieval algorithm? It looks like a validation paper.

Line 67: I don't understand what "has filtered the Mie scattering at different echoes" means. Please clarify

Line 68: "avoids" would be better than "exempts"

Line 75: the reference for Cornut et al. 2023 is missing

Equations 2.6 and 2.8 appear to be the same

---

## Author Comment (AC1)

**Response to Albert Ansmann' comments**

MS No.: amt-2023-219

MS type: Research article

Title: Aerosol Optical Properties Measurement using the Orbiting High Spectral Resolution Lidar onboard DQ-1 Satellite: Retrieval and Validation

Author: Chenxing Zha, Lingbing Bu, Zhi Li, Qin Wang, Ahmad Mubarak, Pasindu Liyanage, Jiqiao Liu, and Weibiao Chen

We greatly appreciate Dear Dr. Albert Ansmann's valuable time reviewing our research paper and providing feedback/suggestions. The comments are valuable and helpful for improving our paper. We have studied all comments carefully and have made major revisions to our manuscript. We sincerely thank the Editor and the reviewer for their valuable suggestions in improving the previous version of our manuscript.

**Report #1**

The manuscript describes the performance of the Chinese aerosol space lidar DQ-1 HSRL. This manuscript will be probably one of the fundamental papers on DQ-1 HSRL. Therefore, the quality of the manuscript must be improved. Much more information about the quality of the measured signals, about calibration and quality assurance efforts, about cross talk in the different measurement channels needs to be provided.

Major revision is thus recommended.

**Response:** We are grateful to the reviewer for acknowledging our efforts. We have followed your suggestions carefully and revised the manuscript accordingly, we highlighted the revised text with red font. The corrections have been made in the revised manuscript. Thank you for your suggestions.

**## General comments and major points**

**Point 1: What is the meaning of DQ-1 (DQ stands for…?, should be explained…).**

**Response 1:** DQ-1 stands for atmospheric environment monitoring satellite, also called DaQi-1, We have changed the introduction in the abstract. Relevant changes have been made in the revised version of the manuscript at:

Line 13 The Atmospheric Environment Monitoring Satellite (AEMS), also called DaQi-1 (DQ-1), was launched in April 2022.

**Point 2: I would suggest not to use dB to characterize signal-to-noise ratios or the cross talk impact. Just use signal ratios such as P1/P2 instead of Q = 10 log (P1/P2) in dB.**

**Response 2:** Thank you for pointing out this issue. We will accept your suggestion and use P1/P2 to characterize signal-to-noise ratios in this manuscript. The corresponding modifications have been made to the manuscript at:

L19: The results have shown a continuous profile alignment between the two datasets, with DQ-1

describing an improved signal-to-noise ratio.

L220: While the value of CALIPSO's aerosol signal SNR varied from 10 to 40, the SNR of DQ-1's aerosol signal exceeded 40. Additionally, DQ-1 has maintained a high-altitude molecular scattering SNR above 20, whereas CALIPSO's high-altitude molecular scattering signal SNR has a value of less than 20.

Line 550 Figure 4 (c), (d):

[Figure]

**Point 3: In the abstract, R^2 0.803, you mean R^2=0.803?**

**Response 3:** Yes, it is. And we apologize for the negligence of the wording here, we have revised the wording to be clearer. The corresponding revised sentences are:

Line 24: …were correlated and yielded a value of $R^2$ equal to 0.803.

**Point 4: The shortcomings of airborne (and spaceborne) observations could also be mentioned: They deliver snapshot-like observations, compared to ground-based (GB) observations. GB remote sensing may allow to study life cycle processes, evolutions, developments.**

**Response 4:** Thank you for your comments. Considering your suggestion, we have added corresponding modifications to the manuscript regarding the advantages and disadvantages of airborne (and spaceborne) observations to our manuscript. The revised sentences are:

Line 37: There are several advantages of ground-based lidar: easy maintenance of the instrument, accurate results, and long-term observation. The life cycle and evolution process of aerosols can be studied through ground-based lidar systems.

Line 54: Nevertheless, the system's observations are limited by factors such as flight paths and meteorological conditions to prevent prolonged data collection, and low spatiotemporal resolution making it impossible to observe the microscale system.

Line 57: Although satellite observations have the drawbacks of low temporal and spatial resolution, as well as long revisit period, they can obtain global aerosol optical parameters

**Point 5: lines 66-68: The text is confusing, please rephrase.**

**Response 5:** We are very sorry for our incorrect writing and thanks for your suggestion. The revised sentence is:

Line 66: As a new-generation lidar, high spectral resolution lidar avoids the assumption during retrieval steps, resulting in more precise results.

**Point 6: Furthermore, one could mention other activities such as the AEOLUS and the EarthCARE space lidar missions, in support to CALIPSO. line 79: …   to be launched…when?**

**Response 6:** Thank you for pointing out this issue. We have added the introduction of the two space lidar missions and their support to Calipso. We are sorry for this incomplete sentence, this satellite is expected to be launched in 2024, and we have made corresponding revisions to the manuscript. The revised sentence is

Line 75: This satellite is anticipated to be launched in 2024. In addition, the Atmospheric Laser Doppler Instrument (ALADIN) lidar loaded on the AEOLUS provides results of global aerosol optical parameters retrieved by L2A data which had performed the comparison with the ground-based lidar product."

**Point 7: line 112: There are three signal channels: cross-polarized (particle+Rayleigh), co-polarized (particle+Rayleigh), and HSRL channel (Rayleigh). Please add this information on particles and Rayleigh contributions;**

**DQ-1 detects co and cross-polarized Rayleigh and particle backscatter components. We need to know, how large the contribution of these four signal components is in the three measurement channels? Is all the cross talk considered and corrected for in the different product retrievals? How are the transmission and reflection properties of the optical elements in the receiver unit, between the telescope and the detector. What is the contribution of the four signal elements to the detected signal counts in the three channels.**

**All potential cross talk effects must be considered in the retrieval. So, please discuss the contribution of the four backscatter components in the three channels.**

**Response 7:** Thank you for pointing out these important issues. We have added the functions of each optical channel to the manuscript, we also analyzed the proportion of Mie scattering and Rayleigh scattering in each channel.

Figure 1 illustrates the schematic of the three optical channels in the DQ-1 receiving system. The function of high spectral channels is to separate the Mie scattering and Rayleigh scattering in the signal. In this channel, the Mie scattering is mostly filtered out, leaving behind molecular Rayleigh scattering. The ratio of these two scattering signals can be determined by comparing the high spectral channel signal with the parallel channel signal. After filtering, only 0.2% of the Mie scattering signal will remain in the high spectral channel. The remaining Mie scattering signal will be discussed in Response 8.

The function of parallel channels and perpendicular channels is to obtain polarization information of the target. A Polarization Beam Splitter (PBS) separates the perpendicular polarized and parallel polarized signals in the echo signal, and by comparing the ratio of these two channels, the polarization information of the target object can be obtained. For parallel (perpendicular) channels, PBS can filter out most of the perpendicular (parallel) polarized signals, making the proportion of vertical (perpendicular) polarized signal received in PMT less than 0.01%, considering the polarization retrieval results of aerosol signals,

this crosstalk will not affect the retrieval of aerosols (Hair et al., 2008).

[Figure]

Figure 1 Schematic of HSRL receiving system.

The corresponding revision has been made to the manuscript at:

line 113: The parallel and perpendicular channels serve the function of obtaining polarization information of the aerosol, a Polarizing Beam Splitter (PBS) is placed to reduce polarization cross talk. The high spectral channels function to separate Mie scattering and Rayleigh scattering in the signal, obtaining the molecular scattering profile.

**Point 8: line 117: You mention, the aerosol suppression ratio is more than 25 dB. So, we have less than -25 dB, i.e., less than 0.00316). Is that sufficient to derive extinction profiles? In the case of cirrus backscatter and extinction, you may need 5-6 orders of magnitude suppression to obtain high-quality extinction profiles? But maybe you correct for cross talk? But that should then be explicitly described.**

**Response 8:** Yes, it is sufficient for retrieval. We present the attenuated backscatter signal profiles of cirrus and comparison using molecular backscattering profiles. The filter absorbed the Mie scattering signal of the cirrus, and the signal quality did not affect the results. In our retrieval results of cirrus, the lidar ratio is less than 40, which is consistent with the range of ice crystals in the literature. Other types of aerosols are also consistent with the literature.

[Figure]

Figure 2 Attenuated backscatter of a cirrus

Figure 2 showcases a cirrus profile. The red line represents the unfiltered signal, the green line represents the filtered signal, and the blue line represents the molecular backscatter signal. The

unfiltered signal displays the echo signal of the cirrus at an altitude of 13-15 km. The filtered signal only contains molecular backscatter, without the cirrus signal. Similarly, clouds at 5-6 km are only displayed in the unfiltered signal. At an altitude of 3 km, the signal is unable to penetrate thick cumulus clouds, resulting in signal attenuation below this altitude. The filtered signal is more in line with the molecular scattering, the signals from clouds and aerosols have been filtered out.

**Point 9: The Eqs.(2.1)-(2.3) are too simple. No efficiency factor (describing optical and detection efficiency), no cross talk factor. So, Eq.(2.5) does not need any calibration? The depolarization ratio is simply obtained from the ratio of the cross- to- co-polarized channel outputs?**

**Response 9:** We apologize for our negligence. System calibration has been conducted in equations (2.1) – (2.3). We have added and modified the corresponding introduction and formulas to the manuscript, analyzing the system calibration process of signals. The corresponding modifications are:

Line 139:

$$B^{\perp}(r) = \frac{P(r)r^2}{P_0\eta^{\perp}AL}[\beta_m^{\perp}(r) + \beta_a^{\perp}(r)] \times exp\left\{-2\int_0^r [\alpha_m(r) + \alpha_a(r)]\, dr\right\} \tag{2.1}$$

$$B_C^{\parallel}(r) = \frac{P(r)r^2}{P_0\eta_C^{\parallel}AL}[\beta_m^{\parallel}(r) + \beta_a^{\parallel}(r)] \times exp\left\{-2\int_0^r [\alpha_m(r) + \alpha_a(r)]\, dr\right\} \tag{2.2}$$

$$B_H^{\parallel}(r) = \frac{P(r)r^2}{P_0\eta_H^{\parallel}AL}[T_m(r)\beta_m^{\parallel}(r) + T_a(r)\beta_a^{\parallel}(r)] \times exp\left\{-2\int_0^r [\alpha_m(r) + \alpha_a(r)]\, dr\right\} \tag{2.3}$$

Line 144: $P(r)$ represents the power of the laser echo signal at distance $r$. $P_0$ represents the pulse energy, $\eta$ represents the optical-electrical efficiency of the recever in corresponding channel, $A$ represents the aperture of the telescope, and $L$ stands for half of the pulse spatial length, where $L$ is calculated as $L = c\Delta t/2$, with c representing the speed of light and $\Delta t$ denoting the pulse duration. System correction has been implemented to ensure that the data is solely contingent upon atmospheric conditions. $\beta_m(r)$ and $\beta_a(r)$ represents the backscatter coefficient of molecules and aerosols respectively. The molecular backscatter coefficient and extinction coefficient are calculated by the S6 molecular model using the data of temperature and pressure provided by ERA5.

**Point 9: An Eq.( ) for T_m would be nice. Please state in the manuscript where you found Eq.(2.4). Please provide a reference for Eq.(2.6) and Eq.(2.8). Please, provide reference for Eq.(2.9). What is the solution for Eq.(2.9) if you start from Eq.(2.8) (or Eq.(2.6))?**

**Response 9:** Following your comment, the formula for T_m has been incorporated into the manuscript. The references and the solution of the corresponding formulas have been added:

Line 145:

$$T_m(T,p) = \int F(v) \int R_m(v',T,p)l(v - v')dv'dv \tag{2.4}$$

$$T_a(T,p) = \int F(v) \int R_a(v',T,p)l(v - v')dv'dv \tag{2.5}$$

Line 147: Where $l(v - v')$ represents the spectrum distribution of the laser beam, $F(v)$ represents the normalized transmission spectrum of the iodine filter. $R_m(v',T,p)$ represents the normalized molecular scattering spectrum related to temperature and pressure. $R_a(v',T,p)$ represents the normalized aerosol

particles scattering spectrum.

Line 148:

$$\beta_a(r) = \beta_m(r)\frac{[1 + \delta(r)]}{(1 + \delta_m)}\frac{[T_m(r) - T_a(r)]K(r)}{[1 - T_a(r)K(r)]} - \beta_m(r), (Xu\ et\ al., 2020) \qquad (2.6)$$

Line 153:

$$\tau(r_0) = \int_0^{r_0}(\alpha_a(r) + \alpha_m(r))\,dr = -\frac{1}{2}ln\left[\frac{(1 - K(r_0)T_a(r_0))(1 + \delta_m)B_H^{\parallel}}{(T_m(r_0) - T_a(r_0))}\right], (Xu\ et\ al., 2020) \quad (2.9)$$

Line 159:

$$\alpha_a(r_0) = \frac{\partial\tau(r_0)}{\partial r} - \alpha_m(r_0) = -\frac{1}{2}\frac{\partial}{\partial r}\left\{ln\left[\frac{(1 - K(r_0)T_a(r_0))(1 + \delta_m)B_H^{\parallel}}{(T_m(r_0) - T_a(r_0))}\right]\right\} - \alpha_m(r_0), (Xu\ et\ al.,2020) \quad (2.10)$$

**Point 10: Please explain all abbreviations when they appear for the first time.**

**Response 10:** We apologize for this carelessness. All abbreviations in the manuscript have been double checked and explained. Revisions are:

Line 21: Optical property profiles from National Aeronautics and Space Administration (NASA) Micro Pulse Lidar NETwork (MPLNET) stations were selected for validation with the DQ-1 measurements

Line 176: The network utilizes the automated sun photometers produced by Cimel Electronique as the primary instrument to observe the atmosphere.

**Point 11: line 184: Pappalardo et al. (2014) deal with the EARLINET lidar network, not with the MPLNET. line 191: For HYSPLIT we need a references.**

**Response 11:** We apologize for this oversight, the corresponding reference has been replaced or added. The revised references are:

Line 184: The NASA Micro-Pulse Lidar Network (MPLNET) is a globally distributed network equipped with a polarized Micro-Pulse Lidar (MPL) system that has been operating continuously since 2000 (Welton et al., 2001).

Line 191: The NOAA Hybrid Single Particle Lagrangian Integrated Trajectory (HYSPLIT) model is a widely used atmospheric transport and dispersion model developed by the National Oceanic and Atmospheric Administration (NOAA) (Draxler and Hess, 1997; Stein et al., 2015).

**Point 12: line 227:    … with a 10^-4 m^-1 values …. What does it mean? What do you want to tell us? line 228:    … with a value of 10^-5 m^-1 sr^-1. …. What does it mean? What do you want to tell us? lines 222 and 234: the same … I do not understand.**

**Response 12:** By representing specific values of aerosol optical parameters, the meaning of these values is to introduce the concentration and types of aerosols here, which helps to identify and analyze.

**Point 13: line 245: CALIPSO does not measure the lidar ratio. The lidar ratio is an input value in the CALIPSO data analysis. The lidar ratio has to be set. line 247: In case of CALIPSO measurements of marine particles the lidar ratio is set to 20 sr. What about the measured lidar ratios (DQ-1 HSRL), measured within the cirrus at 15 km height? Please discuss the values. This is one of the important new products of a spaceborne HSRL, compared to the CALIPSO products. Should be highlighted.**

**Response 13:** We apologize for the oversight regarding the CALIPSO lidar ratio. We have acknowledged this issue in the relevant section of the article. The lidar ratio of DQ-1 has been compared only to CALIPSO to a limited extent. Besides, we have added an analysis of the lidar of cirrus, showcasing highlights of DQ-1. The revised sentences are:

Line 244: The advantage of the DQ-1 HSRL system is that it can retrieval the lidar ratio without assumptions, which is significantly different from CALIPSO. DQ-1 indicate that the lidar ratio of aerosol particles is around 40 sr, describing the characteristics of mixed dust aerosols, consistent with Calipso's aerosol type. For cirrus at altitude of 10 to 15 km, the retrieved lidar ratio of DQ-1 is less than 40 sr, indicating the characteristics of ice crystals. Due to the proximity of the CALIPSO orbit to oceanic regions, below 3 km altitude and at a latitude below 18°N, CALIPSO identified sea salt aerosols, with numerical values representing a lidar ratio of 20 sr.

**Point 14: The lidar ratio dimension is sr and not sr^-1.**

**Response 14:** Thank you for pointing this out and we are sorry about our mistake. The corresponding dimension has been corrected to "sr" in the manuscript.

**Point 15: Regarding cirrus and specular reflection. Is the lidar zenith or off-zenith pointing?**

**Response 15:** lidar is off-zenith pointing at a specific angle. We have added the corresponding introduction about off-zenith pointing in the manuscript:

Line 108: The laser is with two distinct pulses, pulse A and pulse B, to observe the atmosphere practically, the laser beam is off-zenith pointing at a specific angle.

**Point 16: line 313: You mean top height of 8 km or base height of 8 km? Please, specify.**

**Response 16:** We're sorry for the wording issue here. The meaning we want to express is the top height of the dust. The revised sentence is:

line 313: DQ-1 observed a top height of 8 km for the aerosol layer

**Point 17: Figure 1: One should indicate the different channels (B, B_C, B_H)**

**Response 17:** we have added this information to the title of figure 1, the revised title are:

Line 535: The actual measured transmittance spectra of the on-board iodine vapor filter of the DQ-1 satellite. The subfigure in the lower right corner displays the transmittance spectrum in 1110 line. The red solid line delineates the spectral of the echo signal prior to filter (parallel channel), the blue solid line delineates the spectral of the echo signal after the filter (high spectral channel).

**Point 18: Figure 4e: Please mention that mean profiles for 20-22°N (longitudes…?) are shown.**

**Response 18:** Thank you for pointing out this issue. We have mentioned the latitude and longitude range of the mean profiles. The revised title is:

Line 563: (i) Comparison of particulate depolarization mean profile of 20°N to 22°N; (j) Comparison of aerosol backscatter coefficient mean profile of 20°N to 22°N. (k) Comparison of aerosol extinction coefficient mean profile of 20°N to 22°N. (h) Aerosol lidar ratio mean profile of 20°N to 22°N.

**Point 19: Figure 5: the color range should be improved. We have mostly dark blue, sometimes light blue (or cyan) and sometimes yellow. Figure 5: The advantage of DQ-1 HSRL is that lidar ratios can be measured. But lidar ratio results are not shown in Figure 5 (i:backscatter, j: extinction, why not k: lidar ratio). and also, what about depolarization ratio results?**

**Response 19:** Thank you for pointing out this issue. We have changed the colormap used in the manuscript to display data features more clearly. And we have considered your comments and added the results of the corresponding lidar ratio to the depolarization ratio in Figure 5. The modified figures are:

Figure 5e:

[Figure]

Figure 5i, 5h:

[Figure]

**Point 20: Figure 6: MPL retrievals can only deliver particle backscatter profiles. If extinction profiles obtained with MPL and DQ-1 HSRL match then this is just the hint that the lidar ratio was around 50 sr, as assumed in the MPL retrieval. What about depolarization profiles. Figure 7: Again, only backscatter can be compared.**

**Response:** Thank you for pointing out this issue. We have removed the results comparing with the extinction coefficient of MPL and replaced them with the results comparing with the depolarization ratio in Figures 6 and 7. The modified figures are:

Figure 6a, 6d ,and 6g:

[Figure]

Figure 7:

[Figure]

The above is the complete response to your comments. We look forward to hearing from you regarding our responses. We would be glad to respond to any further questions and comments you may have.

**References**

Hair, J. W., Hostetler, C. A., Cook, A. L., Harper, D. B., Ferrare, R. A., Mack, T. L., Welch, W., Izquierdo, L. R., and Hovis, F. E.: Airborne High Spectral Resolution Lidar for profiling aerosol optical properties, Appl. Opt., 47, 6734-6752, 10.1364/AO.47.006734, 2008.

---

## Author Comment (AC2)

**Response to Anonymous Referee #1**

MS No.: amt-2023-219
MS type: Research article

Title: Aerosol Optical Properties Measurement using the Orbiting High Spectral Resolution Lidar onboard DQ-1 Satellite: Retrieval and Validation

Author: Chenxing Zha, Lingbing Bu, Zhi Li, Qin Wang, Ahmad Mubarak, Pasindu Liyanage, Jiqiao Liu, and Weibiao Chen

Dear Professors:

We express our sincere gratitude to the anonymous referee for dedicating his/her valuable time to scrutinize our manuscript and offer constructive feedback and suggestions. The insightful comments have proven instrumental in enhancing the overall quality of our manuscript. Every comment has been meticulously considered, leading us to implement substantial revisions in our manuscript. We also asked native English colleagues to double-check our language. Our heartfelt thanks extend to both the Editor and the referee for their invaluable suggestions, which significantly contributed to the refinement of the earlier version of our manuscript. Hopefully, the revised manuscript will be considered to be published in Atmospheric Measurement Techniques.

**General comments and major points**

**Point 1: Abstract: DQ-1 consists of more than just the lidar**
**Response 1:** We apologize for our wording here. In addition to Aerosol and Carbon Detection Lidar (ACDL), the satellite also includes Wide Spectral Imager (WSI), Participate Observing Scanning Polarimeter (POSP), Environment Monitoring Instrument-2 (EMI-2), and Directional Polarization Camera-2 (DPC-2). the HSRL system is one part of ACDL. Corresponding modifications have been made to the manuscript, and the corresponding changes are displayed in red font.

Line 13: The Atmospheric Environment Monitoring Satellite (AEMS), also called DaQi-1 (DQ-1), was launched in April 2022, one of its main payloads is the High Spectral Resolution Lidar (HSRL) system.

**Point 2: Abstract: Linear depolarization ratio?**
**Response 2:** Thank you for bringing this matter to our attention. DQ-1 retrieved the linear polarization of the aerosol, and we have made modifications to the terms in the abstract at:

Line 17: This method has retrieved the aerosol linear depolarization ratio, backscatter coefficient, extinction coefficient, and optical depth.

**Point 3: l. 30: Probably $10^2$ is meant here. 10 nm as maximum is certainly wrong. And in principle there is no lower limit until you reach the molecular level at about 0.1 nm.**
**Response 3:** We apologize for this significant error, the value has been corrected at:

Line 30: Aerosols are tiny solid and liquid particles suspended in the atmosphere, an atmospheric aerosol particle typically ranges from 0.01 to 10 μm in diameter.

**Point 4: l. 38: I would not assume results from ground-based lidar systems to be more accurate in general. Especially when e.g. assuming daytime measurements with Fernald/Klett algorithms and no further information of the lidar ratio or the long integration time needed for night-time Raman measurements.**

**Response 4:** Thank you for pointing out this issue. Due to the use of traditional Fernald methods for retrieval in ground-based lidars such as MPLNET, it should not be stated here that their observations are accurate. Our comparison with MPLNET is qualitative analysis. The sentences have been modified at:

Line 38: There are several advantages of ground-based lidar: easy maintenance of the instrument, long-term stable observation of specific areas (Mattis et al., 2004; Pitari et al., 2013).

**Point 5: l. 39: Also compared to what; airborne or spaceborne measurements? 'Additionally, comparing…' This sentence seems to be at the wrong place here.**

**Response 5:** We are sorry about our wording here. What we want to express is that ground-based data can be compared with spaceborne and airborne data.

Line 39: Furthermore, it is beneficial to validate spaceborne and airborne measurements with ground-based lidar systems.

**Point 6: l. 65: Was fuel consumption really the reason? I also heard of solar panel degradation. Maybe you should add a reference.**

**Response 6:** The source of this information is the Calipso webpage of LARC (https://www-calipso.larc.nasa.gov/), its original text is:" *Fuel reserves are now exhausted*, and in its decaying orbit the satellite can no longer generate sufficient power to operate the science instruments." We have added this reference to the manuscript at:

Line 65: Due to fuel consumption (Langley Research Center, 2024), CALIPSO was retired in August 2023, a well-established and developed new-generation spaceborne Lidar is needed to replace CALIPSO for global aerosol observation.

*The Cloud-Aerosol Lidar and Infrared Pathfinder Satellite Observation (CALIPSO): https://www-calipso.larc.nasa.gov/, last access: Jan 2nd.*

**Point 7: l. 73: According to the latest developments AOS will not incorporate an HSRL channel. But this may change again if NASA sees the benefit of the DQ-1 data.**

**Response 7:** Thank you for pointing out this issue. According to the literature cited in this sentence, AOS program have studied the scientific value of spaceborne HSRL systems. We have modified the manuscript, the corresponding changes are displayed in red font.

Line 73: Furthermore, under the leadership of NASA, the Atmosphere Observing System (AOS) international program analyzes the additional value provided by the spaceborne HSRL system. This research has shown that the results of spaceborne HSRL systems are more accurate than the results of traditional elastic backscatter lidar in three different cases (Cornut et al., 2023).

**Point 8: l. 83: Does duel-polarization mean duel wavelength?**

**Response 8:** The DQ-1 HSRL system only includes lasers with a wavelength of 532.245 nm. The meaning of "dual" refers to parallel polarization and perpendicular polarization.

**Point 9: l. 108: To which line catalogue corresponds this numbering? Please give a reference.**

**Response 9:** We apologize for our negligence, the reference has been added to the manuscript at:

Line 108: The laser beam of the HSRL system has a wavelength of 532.245 nm, a pulse repetition frequency of 40 Hz, and the absorption line of iodine molecules corresponds to line 1110 (Weibiao et al., 2023).

*Weibiao, C., Jiqiao, L., Xia, H., Huaguo, Z., Xiuhua, M., Yuan, W., and Xiaopeng, Z.: Lidar Technology for Atmosphere Environment Monitoring Satellite, Aerospace Shanghai (Chinese & English), 40, 13-20, 10.19328/j.cnki.2096-8655.2023.03.002, 2023.*

**Point 10: l. 110: What exactly is meant by 'categorized' and 'adjusted' here?**

**Response 10:** We are sorry about our wording and mistake here. What we want to express is that the purpose of our processing is to normalize these two pulses. Corresponding revision has been made to the manuscript at:

Line 110: Both of the pulses are normalized prior to the retrieval process.

**Point 11: l. 118: 25 db is quite low for an Iodine filter. Is this limited by the iodine vapor pressure or is there some spectral impurity of the laser? And how stable is this value, as it has to be considered in the retrieval?**

**Response 11:** Thank you for raising this important question. The transmittance of the iodine filter is related to its temperature and pressure. The temperature and pressure of the iodine filter on the DQ-1 HSRL are controlled within a certain range to ensure stable transmittance.

25 db is sufficient to filter out the Mie scattering for subsequent retrieval. To illustrate this issue, we have selected a representative profile to analyze its filtering effect. Figure 1 showcases an attenuated backscatter profile. The red line represents the unfiltered signal, the green line represents the filtered signal, and the blue line represents the molecular backscatter signal. The unfiltered signal displays the echo signal of the cirrus at an altitude of 13-15 km. The signals of cirrus contain strong Mie scattering signals, the filtered signal only contains molecular backscatter, without the Mie scattering signal. Similarly, clouds at 5-6 km are only observed in the unfiltered signal. At an altitude of 3 km, the signal cannot penetrate thick cumulus clouds, resulting in signal attenuation below this altitude. The filtered signal is more in line with the molecular scattering, the signals from clouds and aerosols have been filtered out.

[Figure]

Figure 1 Attenuated backscatter profile

The stability of this value is related to the stability of the laser frequency. Before the satellite was launched, the stability of the seed laser was tested, and the results are shown in Figure 2, the standard deviation of frequency variation within 3 hours is 0.8 MHz@rms (Weibiao et al., 2023). The stability of the frequency can also ensure the stability of the suppression ratio of the iodine filter (Dong et al., 2018).

[Figure]

Fugure 2 Frequency stabilization measurement by the 1064 nm seeder laser

**Point 12: l.125: 'A and B…' It is not exactly clear what is meant here with interleaved.**

**Response 12:** We are sorry about our mistake here. The sentences here are revised to:

Line 125: To improve the signal-to-noise ratio of the original data, the two pulses have been normalized and averaged.

**Point 13: l. 125: compare = achieve?**

**Response 13:** We apologize for our previous wording choice. 'achieve' is a more accurate verb to use.

Line 125: To achieve the design's horizontal resolution of 20 km, we used 48 sets to configure the pulse averaging iterations, and the vertical resolution is 48 meters.

**Point 14: P.6, Eq 2.8: This is a replication of Eq. 2.6. Please delete.**

**Response 14:** We apologize for our mistake. We have deleted the incorrect formula. All formulas have been rechecked.

**Point 15: l. 236f: Why should this be a mixture, here directly of the desert? In the retrieval the dust plume looks extremely homogenous, with no sign of mixing; And 50 sr does not really point to mixed dust.**

**Response 15:** According to the literature, due to the aerosol depolarization ratio observed by DQ-1 being 0.2, it belongs to the range of dust. Considering your comments, we will no longer refer to it as a mixed dust.

[Figure]

Figure 3 Classification model of HSRL (Burton et al., 2012).

[Figure]

Figure 4 Aerosol depolarization ratio mean profile of 20°N to 22°N measured by DQ-1 on June 6, 2022.

The corresponding modifications were made to the manuscript at:

Line 236: The value of the depolarization ratio at low altitude obtained from DQ-1 is 0.2, demonstrating the nature of dust.

Line 244: The advantage of the DQ-1 HSRL system is that it can retrieval the lidar ratio without assumptions, which is significantly different from CALIPSO. DQ-1 indicate that the lidar ratio of aerosol particles is around 40 sr, describing the characteristics of aerosols, consistent with Calipso's aerosol type.

Line 317: On July 3rd, measurements over West Africa indicated a lidar ratio distribution centered

around 50 sr and depolarization ratio values within the range of 0.3 to 0.4, representing dust.

**Point 16: l. 243ff: What is meant here by laser energy attenuation? A smaller laser energy does not alter the mean profile.**

**Response 16:** Due to CALIPSO's prolonged in orbit operation, laser energy attenuation causes diminished signal-to-noise ratio, leading to more noise signals within the echo signal and consequently, increased measurement inaccuracies. As depicted in Figure 5, the SNR profile of DQ-1 is shown within a spatial region identical to that of CALIPSO. Within the latitude range of 40°N to 10°N, the signal-to-noise ratio of DQ-1 measures 60, while CALIPSO measures a lower value of 20. The noise signal of CALIPSO affecting the retrieval results, resulting in the depolarization ratio results exceeding the credible range.

[Figure]

Figure 5 SNR of DQ-1 and CALIPSO total attenuated backscatter

**Point 17: l.245: For CALIPSO this value comes from a database and is not measured.**

**Response 17:** We apologize for our negligence. We have acknowledged this issue in the relevant section of the article. Due to the lack of accurate lidar ratio results at present, we chose to compare with CALIPSO. The lidar ratio of DQ-1 was only qualitatively compared with CALIPSO. The corresponding modifications were made to the manuscript at:

Line 244: The advantage of the DQ-1 HSRL system is that it can retrieval the lidar ratio without assumptions, which is significantly different from CALIPSO. DQ-1 indicate that the lidar ratio of aerosol particles is around 40 sr, describing the characteristics of dust aerosols, consistent with Calipso's aerosol type.

**Point 18: l. 256: The sentence says the same as the previous one.**

**Response 18:** We apologize for our mistake. The corresponding modification has been made to manuscript at:

Line 254: This comparison has three surface types: land, ocean, and coastal regions. Profile-averaged findings inside a circle with a radius of 100 kilometers, centered on MPLNET stations, were explicitly chosen for DQ-1 data. For MPLNET data, we utilized the average MPLNET profile within 15 minutes of DQ-1 transit.

**Point 19: l. 319: Particle depolarization ratio would be much more valuable;**

**Figure 9: What sort of depolarization ratio is shown? According to the main text it is volume depolarization but particle depolarization would be more interesting.**

**l., 320: If this is volume depolarization ratio, like stated above, this is not necessarily the case. The constant lidar ratio does not indicate a large amount of mixing. Please provide aerosol particle depolarization. Maybe in the instrument section a paragraph should be added on how the depolarization measurements are calibrated and how large a possible instrument related depolarization is.**

**Response 19:** Thank you for pointing out this issue. We have changed to using the particulate depolarization ratio to analyze the African dust instead of the volume depolarization ratio. After considering your opinion and our discussion, we also believe that this does not reflect the characteristics of mixed dust.

On the issue of the possible instrument related depolarization. The clean atmosphere at high altitudes only contains atmospheric molecules. Retrieving the depolarization ratio of atmospheric molecules at high altitudes can determine whether there are possible instrument related errors. We have chosen a profile illustrating a high-altitude depolarization ratio result, as depicted in Figure 6. The results show that DQ-1 retrieved the molecular depolarization ratio of 1%, which is consistent with the literature (Young, 1982; Rowell et al., 1971), so we believe that the instruments are insufficient to cause errors in the retrieval results.

[Figure]

Figure 6 Profile of DQ-1 depolarization ratio

The system calibration regarding the depolarization ratio has been added. The corresponding modification has been made to the manuscript at:

Line 139:

$$B^{\perp}(r) = \frac{P(r)r^2}{P_0 \eta^{\perp} AL} [\beta_m^{\perp}(r) + \beta_a^{\perp}(r)] \times exp\left\{ -2\int_0^r [\alpha_m(r) + \alpha_a(r)]\, dr \right\} \tag{2.1}$$

$$B_C^{\parallel}(r) = \frac{P(r)r^2}{P_0 \eta_C^{\parallel} AL} [\beta_m^{\parallel}(r) + \beta_a^{\parallel}(r)] \times exp\left\{ -2\int_0^r [\alpha_m(r) + \alpha_a(r)]\, dr \right\} \tag{2.2}$$

$$B_H^{\parallel}(r) = \frac{P(r)r^2}{P_0 \eta_H^{\parallel} AL} [T_m(r)\beta_m^{\parallel}(r) + T_a(r)\beta_a^{\parallel}(r)] \times exp\left\{ -2\int_0^r [\alpha_m(r) + \alpha_a(r)]\, dr \right\} \tag{2.3}$$

Line 144: $P(r)$ represents the power of the laser echo signal at distance $r$. $P_0$ represents the emitting power of the laser, $\eta$ represents the optical efficiency of the corresponding receiving channel, $A$ represents the aperture of the telescope, and $L$ stands for the half of the pulse spatial transfer length, where $L$ is calculated as $L = c\Delta t/2$, with $c$ representing the speed of light and $\Delta t$ denoting the pulse duration. System correction has been implemented to ensure that the data is solely contingent upon atmospheric conditions. $\beta_m(r)$ and $\beta_a(r)$ represents the backscatter coefficient of molecules and aerosols respectively, $\alpha_m(r)$ and $\alpha_a(r)$ represents the molecular and the aerosol extinction coefficients. The molecular backscatter coefficient and extinction coefficient are calculated by the S6 molecular model using the data of temperature and pressure provided by ERA5.

Line 317: On July 3rd, measurements over West Africa indicated a lidar ratio distribution centered around 50 sr and the particulate depolarization ratio values within the range of 0.25 to 0.4, representing dust.

Line 319: During the transport process, the value of the lidar ratio was constant at 50 sr, while the particulate depolarization ratio was reduced from 0.25 to 0.15.

Line 325: DQ-1 observed that as these aerosols were transported, their altitude, backscatter coefficient, and particulate depolarization ratio decreased at the lidar ratio's constant value.

Line 593 Figure 9c:

[Figure]

**Point 20: l.333: The backscatter coefficient does not depend on laser energy, only the raw signals. So, there is nothing to correct.**

Response 20: Thank you for pointing out this issue and we are sorry for our wording here. The correction we made refers to the correction based on molecular scattering at the reference height, and it is inappropriate to refer it to energy correction. The corresponding modification has been made to the manuscript at:

Line 331: As DQ-1 data is available from June to December 2022, we substituted data from February to June 2022 with CALIPSO data. Figure 10 presents the observed attenuated backscatter coefficient

from January to December 2022 within the stratosphere over the South Atlantic Ocean, using both CALIPSO and DQ-1.

**Point 21: P.12, l. 344: I would recommend to talk of volcanic aerosol not volcanic ash. The ash sediments out quite fast. What goes into the stratosphere are mostly condensated gases like H₂O or H₂SO₄ etc.**

**Response 21:** Thank you for pointing out this issue. We accepted your comment and have made relevant modifications at:

Line 25: We use the DQ-1 dataset to initially investigate the transport processes of the Saharan dust and the South Atlantic volcanic aerosols.

Line 98: We use data from DQ-1 to analyze the transport processes of Saharan dust and South Atlantic tropospheric volcanic aerosol.

Line 330: DQ-1 observations over the South Atlantic have revealed various effects and distribution of volcanic aerosol in the stratosphere.

Line 340: By May 1st, due to insufficient laser energy, CALIPSO received weak volcanic aerosol backscatter signals, making them difficult to distinguish from system noise.

Line 343: Due to the diffusion of volcanic aerosols in the stratosphere and the advantages in laser energy of the DQ-1 system, the results from DQ-1 indicate a broader distribution range of volcanic aerosols. By August 1st, volcanic aerosols had extended southward to 50° S latitude, with an altitude of less than 20 km.

**Point 22: P.13, l. 353: It has not been shown that the outcome of the depolarization ratio from DQ-1 is more reliable. No analyses of the systematical error of the depolarization measurements have been provided.**

**Response 22:** We apologize for this oversight. The reliability of DQ-1's depolarization ratio depends on the analysis of system errors. The corresponding modification has been made to the manuscript at:

Line 353: The aerosol optical parameters obtained from DQ-1 has been validated against the product of CALIPSO and molecular backscatter coefficients. The results indicated that DQ-1 exhibited a higher signal-to-noise ratio and conforms to the results of trends in molecular scattering.

**Point 23: Figure 4: The profile (e) for Calipso seems wrong and also does not fit to the data given in the plot (b). In the profile plot (e) the data of Calipso drops below 10^-5 above 14 km. But this behavior is not present in (b) where ist stays between 10^-3 to 10^-4.**
**l. 215: There seems to something wrong with the CALIPSO backscatter profile. See remark at the plot.**

**Response 23:** We apologize for the mistake. After a comprehensive inspection, the height axis of Figure 4e is incorrect. The current image has been modified correctly, and the corresponding descriptions and analysis in the manuscript has been modified at:

Line 215: The raw signals of the DQ-1 and the CALIPSO align with the molecular scattering profile.

Line 223: In conclusion, the two satellites give consistent raw data results in close orbits, DQ-1 operating at a higher resolution, achieves a better signal-to-noise ratio.

Line 550 Figure 4e:

[Figure]

**Point 24: P.29, Figure 9: The figures are too small to see all details. I would suggest to put them below each other and enlarge them to fill the whole page width.**

**Response 24:** We apologize for neglecting the size of the figures. Corresponding modifications to the figures in:

Line 591: Figure 9a:

[Figure]

Line 591: Figure 9b:

[Figure]

Line 591: Figure 9c:

[Figure]

**Point 25: P.30, Table 1: What is missing here is the optical efficiency of the receiver optics including sun-light filter and the quantum efficiency of the detector. The measurement accuracy depends on the atmosphere probed, i.e. on altitude and aerosol content and also on ambient light conditions. So, one number is not characterizing this well. Furthermore, horizontal resolution would be a better wording than parallel resolution.**

**Response 25:** Thank you very much for your questions and suggestions. We have added the necessary parameters to Table 1, the corresponding modification are made to Table 1 at:

Line 602 Table 1:

**Table 1. Main parameters of the DQ-1 HSRL system (Dong et al., 2018; Weibiao et al., 2023).**

| Parameter | Value |
|---|---|
| Laser Wavelength | 532.245nm |
| Laser Energy | ≥150mj |
| Laser Frequency Stability | 1MHz@10000s |
| Laser repetition frequency | 40 Hz |
| Telescope aperture | 1000 mm |

| | |
|---|---|
| Field of view | 0.2 mrad |
| Broadband bandpass filter | 0.45 nm |
| Narrowband FP filter | 30 pm |
| HSRL filters | iodine vapor filter, 1110 line |
| | aerosol signal suppression ratio ≥ 25 dB |
| Overall optical efficiency (excluding iodine filter) | 0.16 at parallel polarized channel |
| | 0.561 at perpendicular polarized channel |
| | 0.375 at high spectral resolution channel |
| Quantum efficiency of the detector | 40% |
| Measurement accuracy | 15% uncertainty with 20km horizontal resolution |

The above is all our responses to you. Thank you very much for your attention and time, we would be glad to respond to any further questions and comments that you may have.

Yours sincerely,

Chenxing Zha, Lingbing Bu, Zhi Li, Qin Wang, Ahmad Mubarak, Pasindu Liyanage, Jiqiao Liu, and Weibiao Chen.

**Reference**

Burton, S. P., Ferrare, R. A., Hostetler, C. A., Hair, J. W., Rogers, R. R., Obland, M. D., Butler, C. F., Cook, A. L., Harper, D. B., and Froyd, K. D.: Aerosol classification using airborne High Spectral Resolution Lidar measurements – methodology and examples, Atmos. Meas. Tech., 5, 73-98, 10.5194/amt-5-73-2012, 2012.

Dong, J., Liu, J., Bi, D., Ma, X., Zhu, X., Zhu, X., and Chen, W.: Optimal iodine absorption line applied for spaceborne high spectral resolution lidar, Appl. Opt., 57, 5413-5419, 10.1364/AO.57.005413, 2018.

Mattis, I., Ansmann, A., Müller, D., Wandinger, U., and Althausen, D.: Multiyear aerosol observations with dual-wavelength Raman lidar in the framework of EARLINET, Journal of Geophysical Research: Atmospheres, 109, https://doi.org/10.1029/2004JD004600, 2004.

Pitari, G., Di Carlo, P., Coppari, E., De Luca, N., Di Genova, G., Iarlori, M., Pietropaolo, E., Rizi, V., and Tuccella, P.: Aerosol measurements at L'Aquila EARLINET station in central Italy: Impact of local sources and large scale transport resolved by LIDAR, Journal of Atmospheric and Solar-Terrestrial Physics, 92, 116-123, https://doi.org/10.1016/j.jastp.2012.11.004, 2013.

Rowell, R. L., Aval, G. M., and Barrett, J. J.: Rayleigh–Raman Depolarization of Laser Light Scattered by Gases, The Journal of Chemical Physics, 54, 1960-1964, 10.1063/1.1675125, 1971.

Weibiao, C., Jiqiao, L., Xia, H., Huaguo, Z., Xiuhua, M., Yuan, W., and Xiaopeng, Z.: Lidar Technology for Atmosphere Environment Monitoring Satellite, Aerospace Shanghai (Chinese & English), 40, 13-20, 10.19328/j.cnki.2096-8655.2023.03.002, 2023.

Young, A. T.: Rayleigh scattering, Physics Today, 35, 42-48, 10.1063/1.2890003, 1982.

---

## Author Comment (AC3)

**Response to David Winker**

MS No.: amt-2023-219
MS type: Research article
Title: Aerosol Optical Properties Measurement using the Orbiting High Spectral Resolution Lidar onboard DQ-1 Satellite: Retrieval and Validation

Author: Chenxing Zha, Lingbing Bu, Zhi Li, Qin Wang, Ahmad Mubarak, Pasindu Liyanage, Jiqiao Liu, and Weibiao Chen

Dear David Winker:

We would like to extend our deepest appreciation to you for investing your precious time in thoroughly reviewing our manuscript and providing us with constructive feedback and suggestions. The invaluable insights shared have played a crucial role in elevating the overall quality of our work. We have taken each comment into careful consideration, resulting in significant revisions to our manuscript. Our heartfelt gratitude also goes to both the Editor and the referee for their invaluable suggestions, which have greatly contributed to refining the previous version of our manuscript. We sincerely hope the revised manuscript will be considered for publication in Atmospheric Measurement Techniques.

**General comments**

**This paper reports important results from the high spectral resolution lidar flying on the DQ-1 satellite. These are the first HSRL aerosol retrievals from space, so the paper is quite significant. The paper presents initial retrieval results and comparisons with other observations but requires additional details and explanation before it is publishable. To better understand the retrieval algorithm and the nature of the data products, more detailed description of the instrument and the signal processing steps applied are necessary. Several figures require more explanation. Since the paper has a focus on validation, methods use to calibrate the instrument and calibration accuracy should be explained, as well as the expected retrieval accuracy. Expressing ratios in dB is confusing. While common in the radar community, in the lidar community ratios are nearly always expressed as linear ratios rather than in dB. I strongly recommend that linear ratios be used rather than dB. When validating instrument retrievals, one would like to compare with data which is more accurate, or equally accurate. This is a problem for DQ-1 because there may not be any suitable profile data available for comparisons. Discussion of comparisons with MPLnet and CALIPSO in Section 3 should acknowledge that aerosol extinction retrievals from the MPLnet and CALIPSO elastic backscatter lidars have significant uncertainties, which may be larger than the uncertainties of the DQ-1 retrievals. Thus the discrepancies between MPLnet and DQ1 could be entirely due to uncertainties in the MPLnet retrieval and it could be difficult to say what the accuracy of the DQ1 retrieval is. Comparisons of column AOD from AERONET, on the other hand, represent a true validation as AERONET AOD is quite accurate.**

**Responses to general comments:** We are very thankful to you for your kind words and positive feedback about our article. We have followed your suggestions carefully and revised the manuscript accordingly, including instrument calibration and calibration accuracy. The comparison with CALIPSO and MPLNET is qualitative and has been explained in the revised manuscript.

**Detailed comments**

**Section 1. Introduction**

**Point 1: Lines 49-51: What instrument was used to conduct the "observational experiments"? Was this the "airborne scaling system for ACDL" mentioned later? Please clarify. Does "multi-source data" refer to ground-based instruments? What type of instruments provided data?**

**Response 1:** We apologize for our wording and mistakes here. The ACDL airborne scaling system conducted this experiment, we have made corresponding modifications to the wording here. The multi-source data includes CALIPSO, ground-based micro pulse lidar, and sun photometer. The corresponding modifications were made to the manuscript at:

Line 49-51: The Shanghai Institute of Optics and Fine Mechanics (SIOM) of the Chinese Academy of Sciences, in collaboration with Nanjing University of Information Science and Technology (NUIST), Zhejiang University (ZJU), and other institutions, has conducted observational experiments on airborne HSRL system at two distinct geographical locations, Dunhuang and Shanhaiguan. This airborne system is a scaled system of the DQ-1 HSRL. The aerosol optical parameters in these two regions obtained by the HSRL system were validated using CALIPSO, ground-based Micro Pulse Lidar (MPL), and sun photometer.

**Point 2: Line 65: CALIPSO was "retired" in fall 2023 but not because of fuel consumption. CALIPSO science operations were terminated in August 2023 because the satellite's solar arrays could no longer generate enough electrical power to operate the CALIOP lidar.**

**Response 2:** Thank you for bringing this matter to our attention. According to the information provided on the NASA LARC website (*https://www-calipso.larc.nasa.gov/, last access: Jan 2nd.*), Calipso's fuel reserves are exhausted, and in its decaying orbit the satellite can no longer generate sufficient power to operate the science instruments. Considering your comment, we believe that the insufficient energy is the real reason for its retirement. Corresponding modifications are made to the manuscript at:

Line 65: Due to insufficient energy (Langley Research Center, 2024), the CALIPSO science mission ended in August 2023, a well-established and developed new-generation spaceborne Lidar is needed to replace CALIPSO for global aerosol observation.

**Point 3: Lines 72-74: Plans for AOS no longer include an HSRL instrument. AOS information on the vertical motion of clouds will come from Doppler radar.**

**Response 3:** We apologize for our negligence, According to the literature cited in this sentence (Cornut et al., 2023), the AOS program has studied the scientific value of spaceborne HSRL systems. We have modified the manuscript at:

Line 73: Furthermore, under the leadership of NASA, the Atmosphere Observing System (AOS) international program analyzes the additional value provided by the spaceborne HSRL system. This research has shown that the results of spaceborne HSRL systems are more accurate than the results of traditional elastic backscatter lidar in three different cases (Cornut et al., 2023).

**Point 4: Line 79: EarthCARE is anticipated to be launched in spring 2024.**

**Response 4:** Thank you for pointing out this issue. We have modified the manuscript at:

Line 38: This satellite is anticipated to be launched in the spring of 2024.

**Point 5: Line 85: Is the "airborne scaling system for ACDL" an airborne simulator of the DQ-1 HSRL?**

**Response 5:** Yes, like the DQ-1, the airborne ACDL system also includes the HSRL system, the main purpose of the airborne experiment is to verify the reliability and accuracy of the spaceborne system.

**Point 6: Line 95: "to ensure the accuracy" – HSRL retrievals should be more accurate than elastic backscatter lidars such as CALIOP and MPLNET. Intercomparisons are useful but elastic backscatter lidars have significant retrieval errors and are not suited to validate HSRL accuracy. The comparison of DQ-1 AOD with AERONET is more helpful.**

**Response 6:** Thank you for pointing out this issue. We apologize for our negligence in the manuscript. The word "accuracy" is not appropriate here. The comparison between DQ-1 and CALIOP, MPLNET belongs to qualitative comparison. Corresponding modifications are made to the manuscript at:

Line 95: The attenuated backscatter coefficient is first compared with the CALIPSO dataset to ensure the accuracy of instrument calibration. The retrieval results were then compared with the corresponding data products of CALIPSO and NASA MPLNET qualitatively. To ensure accuracy, the retrieved aerosol optical depth was against the corresponding data products of AERONET, where the errors were analyzed.

**Section 2. Instrumentation and Method**

**Point 7: Line 118: Is the suppression of the aerosol signal only 25 dB? Using an iodine filter, the suppression of the aerosol signal can be much greater than 25 dB. Can the authors discuss?**

**Response 7:** Thank you for raising this important issue. For an iodine filter at the spaceborne HSRL system, a higher suppression ratio will reduce signal energy and increase the proportion of system noise, resulting in retrieval errors. Based on the previous simulation, the most suitable iodine filter temperature and pressure were selected as well as the length of the filter, resulting in a suppression ratio of 25 dB (Dong et al., 2018). This suppression ratio is sufficient to filter out the Mie scattering for subsequent retrieval. To illustrate this, we have selected a representative profile to analyze its filtering effect. Figure 2 showcases an attenuated backscatter profile. The red line represents the unfiltered signal, the green line represents the filtered signal, and the blue line represents the molecular backscatter signal. The unfiltered signal displays the echo signal of the cirrus at an altitude of 13-15 km. The signals of cirrus contain strong Mie scattering signals, the filtered signal only contains molecular backscatter, without the Mie scattering signal. Similarly, clouds at 5-6 km are only observed in the unfiltered signal. At an altitude of 3 km, the signal cannot penetrate thick cumulus clouds, resulting in signal attenuation below this altitude. The filtered signal is more in line with the molecular scattering, the signals from clouds and aerosols have been filtered out.

[Figure]

Figure 2 Attenuated backscatter profile

The corresponding modifications and figures were made or added to the manuscript at:

Line 118: Based on the previous simulation, the most suitable iodine filter temperature, pressure and length were selected, resulting in a suppression ratio of 25 dB (Dong et al., 2018), this suppression ratio is sufficient to filter out the Mie scattering for subsequent retrieval. Figure 1 shows the comparison of signals before and after filtering, the filtered signal consists of no aerosol Mie scattering, presenting a residual portion of molecular Rayleigh scattering.

Line 535: Figure 1:

[Figure]

Figure 1: The transmittance spectra of the filter and comparison of signals before and after filtering. (a) The actual measured transmittance spectra of the onboard iodine vapor filter of the DQ-1 satellite, the subfigure in the lower right corner display the transmittance spectrum in the 1110 line. The red solid line delineates the spectral of the echo signal prior to the filter (parallel channel), and the blue solid line delineates the spectral of the echo signal after the filter (high spectral channel). (b) Comparison of signals before and after filtering, the red line represents the unfiltered signal, the green line represents the filtered signal, and the blue line represents the molecular backscatter signal.

**Point 8: Lines 121-122: Some discussion of the pre-processing steps is required. What is meant by "signal to noise ratio control" and what type of moving average and pulse averaging is applied?**

**Line 131: Explain what kind of signal smoothing was applied to the pre-processed data. How is this different from the averaging applied as a pre-processing step?**

**Response 8:** We are sorry for our wording and negligence here. To ensure signal quality, we have deducted signals with a low SNR and used a low-pass filtering algorithm for moving averages of signal profiles before retrieval. Due to the horizontal resolution of 20 km designed for the DQ-1 HSRL system, considering the satellite orbital velocity, the pulse averaging process mainly involves normalizing the profiles within a 20 km horizontal range before averaging them. Corresponding modifications were made to the manuscript at:

Lines 121-122: Prior to L2A data retrieval, some pre-processing steps are taken, including SNR control, moving average, and pulse averaging. SNR control refers to removing the backscatter signal with insufficient SNR, this includes removing the heavy cloud-covered signal and removing erroneous echoes under the surface and signals with poor SNRs, this is achieved by setting an SNR threshold. After this, the low-pass filtering algorithm is used to perform moving average on the profile. To achieve the design's horizontal resolution of 20 km, the profiles within a 20 km horizontal range are normalized and averaged.

**Point 9: Table 1 gives laser energy as "> 150 mJ" but the manuscript says laser pulses A and B have different energies. What is the energy of each pulse, A and B, and how they are averaged together? What is the time delay between these two 532 nm pulses? What does it mean they can be "categorized and adjusted during the retrieval process" (Line 110) The operations described in lines 125-127 are not clear. What is meant by "48 sets".**

**Response 9:** We apologize for our negligence and wording here. It is inaccurate to consider that the pulse energy in Table 1 is greater than 150 mj. The energy of both pulses is greater than 120mj. In orbit, the pulse energy of the two pulses is monitored for calibration, Figure 3 shows the changes in pulse energy. After normalization, the pulses within a horizontal spatial distance of 20km were averaged together.

16.1 μs is the time delay of pulses A and B. The 48 sets mean within a horizontal distance of 20 km, DQ-1 emitted 48 times pulses A and B.

[Figure]

Figure 3 Pulse energy of the two pulses

The corresponding modifications are made to the manuscript at:

Line 110: There is an energy difference between laser pulses A and B, where the L2A data have been calibrated during the production, and the time delay of pulses A and B is 16.1 μs. To improve the SNR of the original data, the two pulses have been normalized and averaged.

Line 605: Table 1. Main parameters of the DQ-1 HSRL system

| Parameter | Value |
|---|---|
| Laser Wavelength | 532.245nm |
| Laser Energy | ≥120mj for pulses A and B |
| Laser Frequency Stability | 1MHz@10000s |
| Laser repetition frequency | 40 Hz |
| Telescope aperture | 1000 mm |
| Field of view | 0.2 mrad |
| Broadband bandpass filter | 0.45 nm |
| Narrowband FP filter | 30 pm |
| HSRL filters | iodine vapor filter, 1110 line |
| | aerosol signal suppression ratio ≥ 25 dB |
| Overall optical efficiency (excluding iodine filter) | 0.16 at parallel polarized channel |
| | 0.561 at perpendicular polarized channel |
| | 0.375 at high spectral resolution channel |
| Quantum efficiency of the detector | 40% |
| Error of retrieval result | 15% uncertainty with 20km horizontal resolution |

**Point 10: How is "measurement accuracy" in Table 1 defined? Is this the random error of the parallel channel signal or does it include calibration errors? Measurement accuracy depends on many factors, including background lighting, altitude, and averaging. Under what conditions is the measurement accuracy of 15% achieved? Is this before or after noise reduction is applied?**

**Response 10:** We are sorry for our wording here. What we want to express here is that the relative error between the retrieved aerosol optical parameters and the true values is less than 15%. The corresponding modifications are made to the Table 1 at:

**Table 1. Main parameters of the DQ-1 HSRL system (Dong et al., 2018; Weibiao et al., 2023).**

| Parameter | Value |
|---|---|
| Laser Wavelength | 532.245nm |
| Laser Energy | ≥120mj for pulses A and B |
| Laser Frequency Stability | 1MHz@10000s |
| Laser repetition frequency | 40 Hz |
| Telescope aperture | 1000 mm |
| Field of view | 0.2 mrad |
| Broadband bandpass filter | 0.45 nm |
| Narrowband FP filter | 30 pm |
| HSRL filters | iodine vapor filter, 1110 line |

| | |
|---|---|
| | aerosol signal suppression ratio ≥ 25 dB |
| Overall optical efficiency (excluding iodine filter) | 0.16 at parallel polarized channel |
| | 0.561 at perpendicular polarized channel |
| | 0.375 at high spectral resolution channel |
| Quantum efficiency of the detector | 40% |
| Error of retrieval result | 15% uncertainty with 20km horizontal resolution |

**Point 11: Important parameters are missing from Table 1. Lidar sensitivity depends on more than laser pulse energy. To gain more understanding of data quality and to better interpret intercomparisons, parameters such as receiver field of view, laser linewidth, bandwidth of the Fabry-Perot etalon, detector type (PMT, APD?, other?), detection scheme (analog or photon counting), and dynamic range of the detection system should be given. What is the view angle of the lidar - pointed at nadir or off-nadir?**

**Response 11:** We are sorry for our negligence here, the corresponding parameters were added to Table 1 and the manuscript at:

Table 1:

| Parameter | Value |
|---|---|
| Laser Wavelength | 532.245nm |
| Laser Energy | ≥120mj for pulses A and B |
| Laser Frequency Stability | 1MHz@10000s |
| Laser repetition frequency | 40 Hz |
| Telescope aperture | 1000 mm |
| Field of view | 0.2 mrad |
| Broadband bandpass filter | 0.45 nm |
| Narrowband FP filter | 30 pm |
| HSRL filters | iodine vapor filter, 1110 line |
| | aerosol signal suppression ratio ≥ 25 dB |
| Overall optical efficiency (excluding iodine filter) | 0.16 at parallel polarized channel |
| | 0.561 at perpendicular polarized channel |
| | 0.375 at high spectral resolution channel |
| Quantum efficiency of the PMT detector | 40% in analog scheme |
| Error of retrieval result | 15% uncertainty with 20km horizontal resolution |

Line 110: The laser is with two distinct pulses, pulse A and pulse B, to observe the atmosphere practically, the laser beam is off-zenith pointing at a specific angle.

**Point 12: Line 124 says "L2A data have been calibrated during production". How is calibration of the three signals accomplished? What is the accuracy of this calibration? What is the accuracy of the ratio of the parallel channel to molecular channel and of the volume depolarization ratio.**
**Line 127: The DQ-1 L2A dataset is an input to the algorithm. Explain what the L2A dataset is**

**and what processing steps are used to produce the L2A.**

**Response 12:** The correction includes energy normalization, distance squared correction, and channel efficiency correction, this will be added in section 2.1.2 of the manuscript. Figure 4 illustrates the result of the calibration, which is the attenuated backscatter coefficient. Due to the absence of aerosol distribution at high altitudes, molecular echo signals can also detect the accuracy of results. The attenuation backscatter coefficients of DQ-1 and CALIPSO are consistent with molecular scattering signals at high altitudes, verifying the accuracy of the calibration.

[Figure]

Figure 4 Attenuated backscatter coefficient of DQ-1 and CALIPSO with molecular backscatter
(the same as the revised Figure 4e in the manuscript)

There has been research on the polarization properties of atmospheric molecules. Retrieving the depolarization ratio of atmospheric molecules at high altitudes can verify the accuracy of the ratio of the parallel channel to the perpendicular channel. We have chosen a profile illustrating a high-altitude depolarization ratio result, as depicted in Figure 5. The results show that DQ-1 retrieved the molecular depolarization ratio of 1%, which is consistent with the literature (Young, 1982; Rowell et al., 1971), so we believe that the depolarization ratio is accurately retrieved.

[Figure]

Figure 5 Profile of DQ-1 depolarization ratio at high altitude.

As shown in the previous Figure 2, it also demonstrates the accuracy of parallel channels and molecular channels. After calibration, the echo signals of both channels match the molecular signals, while the high spectral channel experiences signal attenuation due to passing through the iodine filter.

The DQ-1 L2A dataset includes the attenuation backscatter coefficients of parallel channels, perpendicular channels, and high spectral channels. The parameters and steps used to generate L2A have been added to the manuscript:

Line 139:

$$B^\perp(r) = \frac{P(r)r^2}{P_0\eta^\perp AL}[\beta_m^\perp(r) + \beta_a^\perp(r)] \times exp\left\{-2\int_0^r [\alpha_m(r) + \alpha_a(r)]\,dr\right\} \tag{2.1}$$

$$B_C^\parallel(r) = \frac{P(r)r^2}{P_0\eta_C^\parallel AL}[\beta_m^\parallel(r) + \beta_a^\parallel(r)] \times exp\left\{-2\int_0^r [\alpha_m(r) + \alpha_a(r)]\,dr\right\} \tag{2.2}$$

$$B_H^\parallel(r) = \frac{P(r)r^2}{P_0\eta_H^\parallel AL}[T_m(r)\beta_m^\parallel(r) + T_a(r)\beta_a^\parallel(r)] \times exp\left\{-2\int_0^r [\alpha_m(r) + \alpha_a(r)]\,dr\right\} \tag{2.3}$$

Line 144: $P(r)$ represents the power of the laser echo signal at distance $r$. $P_0$ represents the emitting power of the laser, $\eta$ represents the optical efficiency of the corresponding receiving channel, $A$ represents the aperture of the telescope, and $L$ stands for the half of the pulse spatial transfer length, where $L$ is calculated as $L = c\Delta t/2$, with $c$ representing the speed of light and $\Delta t$ denoting the pulse duration.

**Point 13: Line 131: Explain how SNR is used to control data quality.**

**Response 13:** We are sorry about our wording here. The data quality control of SNR refers to setting a threshold to filter out signals with an SNR lower than the threshold, which filters out pseudo signals below the surface and signals below thick clouds. Corresponding modifications are made to the manuscript at:

Lines 121-122: SNR control refers to removing the backscatter signal with insufficient SNR, this includes removing the heavy cloud-covered signal, and removing erroneous echoes under the surface and signals with poor SNRs, this is achieved by setting a SNR threshold.

**Point 14: The authors should discuss the magnitude of Tm and Ta in equation 2.3 and refer to Fig 1. From Fig 1 it looks like Ta is about 0.002 and Tm is perhaps 40-50%. Since this is a validation paper, the authors should address how Tm and Ta vary on-orbit and whether this variation is a source of retrieval uncertainty.**

**Response 14:** Thank you for raising this important question. On orbit, the changes in Tm and Ta are related to two factors, one is the absorption spectrum of the iodine filter, and the other is the temperature and pressure of the atmosphere. The temperature and pressure of iodine in the DQ-1 filter are strictly controlled, ensuring the stability of the absorption spectrum and thus ensuring the accuracy of the retrieval algorithm. We used ERA5, an authoritative dataset, to ensure the accuracy of the atmospheric temperature and pressure brought into the algorithm. The corresponding discussions were added to the manuscript at:

Line 145:

$T_m(T,p)$ and $T_a(T,p)$ respectively represents the transmittance of the echo signal of molecular and aerosol while passing the iodine filter, they can be expressed as:

$$T_m(T,p) = \int F(v) \int R_m(v',T,p)l(v-v')dv'dv \tag{2.4}$$

$$T_a(T,p) = \int F(v) \int R_a(v',T,p)l(v-v')dv'dv \tag{2.5}$$

Where $l(v-v')$ represents the spectrum distribution of the laser beam, $F(v)$ represents the normalized transmission spectrum of the iodine filter. $R_m(v',T,p)$ represents the normalized molecular scattering spectrum related to temperature and pressure. $R_a(v',T,p)$ represents the normalized aerosol particles scattering spectrum (Dong et al., 2018). To ensure the stability of the absorption spectrum, the temperature and pressure of iodine in the filter are strictly controlled on the orbit.

**Point 15: Line 159: Is the extinction profile really computed from a simple derivative as described in Eq 2.9? This method is extremely sensitive to signal noise.**

**Response 15:** The extinction coefficient results in this manuscript were calculated through derivative calculations. The signal SNR of DQ-1 is high, which can meet the requirements of this calculation method. Moreover, since we have already done the signal smoothing and quality control mentioned above, it can meet the requirements of derivative calculations

**Point 16: Line 169-171: These two sentences are not correct. In the last few years, the CALIOP laser was producing an increasing number of laser pulses with near zero energy. As explained above, science operations were terminated in August 2023 because the orbit had precessed to the east and the satellite's solar arrays could no longer generate enough electrical power to operate the lidar.**

**Response 16:** We are sorry for our mistake here. The corresponding modifications were made to the manuscript at:

Line 169: Due to insufficient power supply, the CALIPSO science mission has ended, demanding the deployment of a new satellite platform to continue global observations of clouds and aerosols.

**Section 3. Validation**

**Point 17: Additional detail is needed on how CALIPSO data is used in the intercomparison with DQ-1. "Energy attenuation of the CALIPSO laser" is mentioned a number of times and pointed to as the source of various discrepancies. It is not clear what is meant by "laser energy attenuation". This term needs to be explained. The manuscript mentions "energy differences" between CALIOP and DQ-1 several times but does not explain what the difference is.**

**Response 17:** Thank you for pointing out this issue and we are sorry for our negligence. Due to CALIPSO's prolonged in-orbit operation, laser energy attenuation causes a diminished SNR, leading to more noise signals within the echo signal and consequently, increased measurement inaccuracies. As depicted in Figure 6, the SNR profile of DQ-1 is shown within a spatial region identical to that of CALIPSO. Within the latitude range of 40°N to 10°N, the SNR of DQ-1 measures 60, while CALIPSO measures a lower value of 20. The noise signal of CALIPSO affects the retrieval results, resulting in the depolarization ratio results exceeding the credible range.

[Figure]

Figure 6 SNR of DQ-1 and CALIPSO total attenuated backscatter

Corresponding explanations about energy attenuation were added in the manuscript:

Line 225: Due to CALIPSO's prolonged in-orbit operation, laser energy attenuation causes a diminished SNR, leading to more noise signals within the echo signal and consequently, increased measurement inaccuracies.

**Point 18: l. 83: Line 207: There are no stratocumulus clouds at 15 km altitude. There are clouds at 15 km at about 15N which attenuate the lidar signal and look like dense cirrus.**

**Response 18:** We are sorry for our mistake here. The lidar ratio of these clouds shows the characteristics of ice crystals, which are cirrus. The correct sentences are made to the manuscript at:

Line 207: At an altitude of 15 km, the distribution of dense cirrus was observed, with the satellite's emitted laser failing to penetrate certain portions of the cloud cover.

**Point 19: Line 218: The method used to estimate SNR needs to be described in more detail. Variability of the signal due to noise must be separated from variability of the atmosphere. Expressing ratios in dB is confusing. While common in the radar community, in the lidar community ratios are nearly always expressed as linear ratios rather than in dB. I strongly recommend that linear ratios be used rather than dB.**

**Response 19:** We are sorry for our negligence here. There is less variation in the high-altitude atmosphere, and the variation of echo signals at high altitudes is caused by noise. To separate the noise in the signal, we selected the high-altitude echo signal as the noise for SNR calculation. To represent SNR more intuitively, the use of dB to represent SNR has been eliminated, and the SNR is directly represented by the ratio of noise to echo signal. Corresponding modifications are made to the manuscript at:

Line 218: The local variations in the high-altitude atmosphere are relatively small, making system noise more easily distinguishable. Using high-altitude echo signals as noise and utilizing the ratio of the noise to echo signals as SNR (SNR). To compare both the signal quality, the SNR of the total attenuated backscattered signals was analyzed, as illustrated in Figures 4c and 4d.

Line 220: While the value of CALIPSO's aerosol signal SNR varied from 10 to 40, the SNR of DQ-1's aerosol signal exceeded 40. Additionally, DQ-1 has maintained a high-altitude molecular scattering SNR

above 20, whereas CALIPSO's high-altitude molecular scattering signal SNR has a value of less than 20.

Line 550 Figure 4 (c), (d):

[Figure]

**Point 20: Line 221: What is the "aerosol signal SNR" and how is it computed? For a backscatter lidar, the SNR of the component of the return signal due only to aerosol scattering doesn't make much sense.**

**Response 20:** We are sorry for our wording here. What we want to express is the SNR of echo signals in areas with aerosol distribution, rather than the SNR of aerosol signals. Corresponding modification is made to the manuscript at:

Line 221: In areas with aerosol distribution, the value of CALIPSO's signal SNR varied from 10 to 40, and the SNR of DQ-1's signal exceeded 40.

**Point 21: Line 243-244: Please explain why the difference in depolarization is attributed to CALIPSO and not to DQ-1? What is meant by "the depolarization ratio retrieved by the laser energy attenuation in CALIPSO."**

**Response 21:** Due to CALIPSO's prolonged in-orbit operation, laser energy attenuation causes diminished SNR, leading to more noise signals within the echo signal and consequently, increased measurement inaccuracies. The noise signal of CALIPSO affects the retrieval results, resulting in the depolarization ratio results exceeding the credible range. The corresponding discussion is added to the manuscript at:

Line 225: Due to CALIPSO's prolonged in-orbit operation, laser energy attenuation causes a diminished SNR, leading to more noise signals within the echo signal and consequently, increased measurement inaccuracies.

**Point 22: Line 245: CALIPSO lidar ratios are estimated, not retrieved, so they do not themselves provide validation of lidar ratios retrieved from DQ-1.**

**Response 22:** We apologize for our negligence. We have acknowledged this issue in the relevant section of the manuscript. Due to the relatively new onboard HSRL system of DQ-1 and the lack of accurate lidar ratio results, we chose to compare it with CALIPSO. The lidar ratio of DQ-1 was only qualitatively compared with CALIPSO. The corresponding modifications were made to the manuscript

at:

Line 244: The advantage of the DQ-1 HSRL system is that it can retrieve the lidar ratio without assumptions, which is significantly different from CALIPSO. DQ-1 indicates that the lidar ratio of aerosol particles is around 40 sr, describing the characteristics of dust aerosols, consistent with CALIPSO's aerosol type.

**Point 23: Line 253: Comparisons with MPLnet set a bound on the accuracy of DQ-1 retrievals but do not "ensure the accuracy of aerosol optical property retrieval using DQ-1" because DQ-1 retrievals should be more accurate than those from MPLnet.**

**Response 23:** Thank you for raising the question and we apologize for our negligence. Due to the use of traditional Fernald methods for retrieval in ground lidars such as MPLNET, which may cause errors due to assumptions, it should not be stated here that the observation results are accurate. We only make qualitative comparisons with MPLNET. The corresponding modifications are made to the manuscript at:

Line 253: We qualitatively compared the aerosol optical parameter data products from the NASA MPLNET ground station with the retrieval results of DQ-1.

**Point 24: Line 333: Please describe how the data was 'corrected for energy variations'. If the attenuated backscatter profiles are properly calibrated, further correction for energy variations should not be necessary.**

**Response 24:** We are sorry about our wording here. The correction we made refers to the correction based on molecular scattering at the reference height, and it is inappropriate to refer to it as energy correction. The corresponding modification has been made to the manuscript at:

Line 331: As DQ-1 data is available from June to December 2022, we substituted data from February to June 2022 with CALIPSO data. Figure 10 presents the observed attenuated backscatter coefficient from January to December 2022 within the stratosphere over the South Atlantic Ocean, using both CALIPSO and DQ-1.

**Point 25: Line 353 states that DQ-1 yields more reliable depolarization ratios than CALIOP. This statement needs more discussion and needs to be supported by analysis. How was it established that DQ-1 is "more reliable", does this mean more accurate?**

**Response 25:** We apologize for our wording here. The reliability of DQ-1's depolarization ratio depends on the analysis of system errors. The expressions "more reliable" or "more accurate" are not appropriate to use here. The corresponding modification has been made to the manuscript at:

Line 353: The aerosol optical parameters obtained from DQ-1 have been validated against the product of CALIPSO and molecular backscatter coefficients. The results indicated that DQ-1 exhibited a higher SNR and conforms to the results of trends in molecular scattering.

**Comments on figures**

**Point 26: There is a problem in the way the CALIOP profile has been plotted in Fig 4e. Does "raw**

signal" refer to the Level 1 attenuated backscatter profiles? Inspection of CALIOP browse images shows the mean 532 nm attenuated backscatter at 20 km is roughly 1E-4 /km/sr whereas Fig 4e indicates the attenuated backscatter is already below 1E-4 /km/sr at 10 km altitude. Please explain how the CALIOP profile in Fig 4e was computed.

**Response 26:** We apologize for the mistake. After a comprehensive inspection and full double-check of the figure in the manuscript, the height axis of Figure 4e is incorrect. The figure has been modified correctly, and the corresponding descriptions and analysis in the manuscript have been modified at:

Line 215: The raw signals of the DQ-1 and the CALIPSO align with the molecular scattering profile.

Line 223: In conclusion, the two satellites give consistent raw data results in close orbits, DQ-1 operating at a higher resolution, achieving a better SNR.

Line 550 Figure 4e:

[Figure]

**Point 27: Figure 8 shows a significant high bias in many of the AOD retrievals from DQ-1. The authors attribute this high bias to cloud contamination. Have the authors demonstrated this? Improved removal of clouds should improve the agreement or might reveal other sources of bias.**

**Response 27:** The results in Figure 8 have already utilized the improved cloud removal algorithm. Before improving the cloud removal algorithm, a smaller correlation coefficient was obtained. The current cloud removal algorithm is based on the backscatter ratio (Ke et al., 2022). but its drawback is that it does not have a good removal effect on the edges of the cloud layer, so this comparison is still partially affected by the cloud layer.

**Point 28: In Section 4.1 it is stated that the backscatter coefficient and volume depolarization of Sahara dust decreased during transport across the Atlantic, while the dust lidar ratio remained constant. This is difficult to tell from Figure 9. It would be helpful to add additional plots which show these trends (or lack of trend) more clearly.**

**Response 28:** Thanks for raising the issue here. We have enlarged the figures and added the figures of mean profile to better show the trends. Corresponding modifications were made to the figures at:

Figure 9:

[Figure]

**Point 29: What do the data curtains in Figure 10 represent? Is each curtain a single orbit (at what longitude?) on the first day of the month, or the average of several orbits (over what range of longitudes) averaged over a month of data?**

**Response 29:** We apologize for our negligence, the corresponding information has been added to the title at:

**Line 599:**

Figure 10 Observed volcanic aerosol attenuated backscatter profile in the stratosphere over the South Atlantic in 2022. The left axis displays date, while the bottom axis displays latitude, with an altitude range of 18 to 26 km, displaying the observation results of a single orbit passing through the central South Pacific (10 ° to 30 ° W) on the first day of each month. The results from January 1st to May 1st are derived from CALIPSO, while the results from June 1st to December 1st are derived from DQ-1.

**Minor issues and technical corrections**

**Point 30: Line 56: "This global information values …" should be "This global information is valuable …".**
**Response 30:** We are sorry for our wording here. The corresponding modification was made to the manuscript at:

Line 56: This global information is valuable for tracking aerosol particle dispersion pathways and compensates for the limitations of ground-based and airborne observations.

**Point 31: Line 64: Does Chiang et al. 2011 describe a retrieval algorithm? It looks like a validation paper.**
**Response 31:** We are sorry for our mistake here. The correct citation position of this literature has been modified to:

Line 61: The Cloud-Aerosol Lidar and Infrared Pathfinder Satellite Observation (CALIPSO), developed by NASA, is the most representative spaceborne lidar satellite. Since its launch in 2006, it has been fully verified by comparing its dataset with other multi-source datasets (Bibi et al., 2015; Wang et al., 2016; Mcgill et al., 2007; Chiang et al., 2011).

**Point 32: Line 67: I don't understand what "has filtered the Mie scattering at different echoes" means. Please clarify. Line 68: "avoids" would be better than "exempts"**
**Response 32:** We are sorry for our wording here. The corresponding modification was made to the manuscript at:

Line 66: As a new-generation lidar, high spectral resolution lidar filters out the Mie scattering in the return signals through a filter. This method avoids the assumptions made by traditional lidar during retrieval, resulting in more precise results.

**Point 33: Line 75: the reference for Cornut et al. 2023 is missing.**
**Response 33:** The reference for Cornut et al. 2023 is on line 407 of the manuscript.

**Point 34: Equations 2.6 and 2.8 appear to be the same**
**Response 34:** We are sorry for our mistake here. The duplicate formula has been deleted and other formulas have been rechecked.

**The above is the complete response to your comments. We look forward to hearing from you regarding our responses. We would be glad to respond to any further questions and comments you may have.**

**Reference**

Cornut, F., Amraoui, L., Cuesta, J., and Blanc, J.: Added Value of Aerosol Observations of a Future AOS High Spectral Resolution Lidar with Respect to Classic Backscatter Spaceborne Lidar Measurements, Remote Sensing, 15, 506, 10.3390/rs15020506, 2023.

Dong, J., Liu, J., Bi, D., Ma, X., Zhu, X., Zhu, X., and Chen, W.: Optimal iodine absorption line applied for spaceborne high spectral resolution lidar, Appl. Opt., 57, 5413-5419, 10.1364/AO.57.005413, 2018.

Rowell, R. L., Aval, G. M., and Barrett, J. J.: Rayleigh–Raman Depolarization of Laser Light Scattered by Gases, The Journal of Chemical Physics, 54, 1960-1964, 10.1063/1.1675125, 1971.

Weibiao, C., Jiqiao, L., Xia, H., Huaguo, Z., Xiuhua, M., Yuan, W., and Xiaopeng, Z.: Lidar Technology for Atmosphere Environment Monitoring Satellite, Aerospace Shanghai (Chinese & English), 40, 13-20, 10.19328/j.cnki.2096-8655.2023.03.002, 2023.

Young, A. T.: Rayleigh scattering, Physics Today, 35, 42-48, 10.1063/1.2890003, 1982.

---

## Referee Report (RR1)

**General comments**

I would like to thank the authors for making significant improvements to the first version of the manuscript. Several corrections and additional details are needed before publication though. Several of my comments below are related to the comparisons between DQ-1 and CALIPSO and the authors' speculation on reasons for differences between the retrievals.

**Specific comments**

For aerosols, Equation 2.8 is a poor approximation of the correct relationship, which has a dependence on scattering ratio. The correct expression can be found in Tesche et al. JGR 2009 (Equation 11). Use of Eqn 2.8 will cause particulate depolarization to be underestimated for aerosols, because the scattering ratios are usually quite low. Cirrus have much higher scattering ratios and Eqn 2.8 can be a useful approximation.

Line 248 – there is a 400 km offset between the two instruments. The lack of cirrus at 15 km in Fig 5d is likely because there was no cirrus in the curtain observed by CALIOP. CALIOP can detect cirrus with extinction even less than 0.01 /km. I am confident the cirrus observed by DQ-1 at 15 km in Fig 5c would have been detected if CALIPSO had been flying in the same orbit as DQ-1.

The particulate depolarization of the dust layer in Fig 5e looks too small. Tesche et al. (2009) found a campaign-average particulate depolarization ratio of 30% for desert dust during the SAMUM campaign in the western Sahara desert. As mentioned above, use of Eq 2.8 to derive particulate depolarization from the measured volume depolarization underestimates the particulate depolarization.

How was the CALIPSO depolarization profile in Figure 5i derived? The data in Fig 5f looks too noisy to produce the smooth profile in Fig 5i. My analysis of the CALIPSO particulate depolarization profile within the dust layer between 24N and 30N showed the profile of depolarization is quite noisy. I found the mean particulate depolarization was 25%.

Line 258 – when I look at this scene, I find some noise spikes giving particulate depolarization as high as 0.5 but mean values are less than 0.3.

Lines 262-264 seem to attribute the difference in particulate depolarization seen in Fig 5e, 5f, and 5i to a difference in SNR between the two instruments. CALIOP and DQ-1 initially had similar 532 nm pulse energy (125 mJ). The pulse energy of the CALIOP laser decreased by 10% to 20% over the life of the mission, which decreased the SNR by 5-10%. This change in SNR would have only a small effect on retrievals and does not explain the differences seen here between DQ-1 and CALIOP data. In the standard CALIPSO data processing of the dust layer in this scene, CALIOP backscatter profile data is only averaged 1 km horizontally – much less than for DQ-1. This is likely the major reason that CALIOP data appears noisier. But the difference between the DQ-1 and CALIOP mean depolarization is probably due to the use of Eqn. 2.8 for DQ-1.

Table 1. It is not clear what 'Retrieval result error' refers to. Retrieval of which product? Under what conditions? Retrieval error usually depends on SNR so varies with altitude and is different for day and night. The text should describe how this error was determined - from validation intercomparisons?

It is mentioned in the author's Response to Reviewers that the HSRL retrieves a molecular depolarization of 1% at high altitudes, but this not mentioned in the manuscript. A broadband measurement of molecular depolarization should give a value somewhat larger than 1%. The DQ-1 HSRL uses a narrowband FP filter with a bandwidth of 30 pm. In this case the molecular depolarization should be about 0.3% because the filter rejects the Raman scattering lines and only passes the central Cabannes line, which is less depolarized. Do the authors know the polarization purity of the transmitted beam?

**Minor comments:**

Line 117 says the laser beam points off-zenith at 'a specific angle'. How far off zenith is the laser pointed and is the angle always the same, or has it changed during the DQ-1 mission?

Line 134-136 – SNR is more difficult to estimate than signal magnitude, how is the threshold for 'SNR control' determined? Is the threshold actually defined in terms of SNR, or in terms of signal magnitude?

Line 162 – please define "S6 molecular model" or provide a reference

Line 252-253 – the authors should be consistent in the units used for backscatter, choose either /km/sr or m/sr and use consistently throughout the paper

**Minor corrections**

Line 17 – 'prominently' is not the right word here; maybe 'clearly', or just say that retrieval algorithms and validation are necessary

Line 20 – the meaning of 'continuous profile alinement' is not clear, does this mean the lidar profiles from the two instruments are similar (when the orbit tracks are close)?

Line 20 – 'showing' rather than 'describing'?

Line 80 – 'a' spaceborne HSRL sounds better here than 'the' spaceborne HSRL

Line 116 – 'The laser produces two …' rather than 'The laser is with two …'

Line 122 – 'high spectral resolution channels' rather than 'high spectral channels' ?

Lines 129-130 – Regarding "the filtered signal consists of no aerosol Mie scattering", I agree there is no significant aerosol signal visible in the filtered signal curve of Fig 1b but strong Mie scattering might result in significant leakage into the filtered signal. From the inset in Figure 1a, it looks like the unfiltered signal peak at 6 km altitude in Fig 1b will enhance the filtered Rayleigh signal by perhaps 20%.

Line 139 – Is 16.1 usec the correct value? According to Dai et al. (2023), the time delay between pulses A and B is 200 usec.

Line 207 – 'religions' should be 'regions'

Line 228 – stratocumulus clouds would not be found at an altitude of 15 km. These are likely some type of cirrus

Line 228 – rather than 'satellite's emitted laser', maybe 'laser return signal'

**REFERENCES**

Dai et al., 2023: Aerosols and Clouds data processing and optical properties retrieval algorithms for the spaceborne ACDL/DQ-1. https://doi.org/10.5194/egusphere-2023-2182

Tesche et al, 2009: "Vertically resolved separation of dust and smoke over Cape Verde using multiwavelength Raman and polarization lidars during Saharan Mineral Dust Experiment 2008" J. Geophys. Res. doi:10.1029/2009JD011862

---

## Author Response (AR2)

**Response to reviewer and editor**

MS No.: amt-2023-219
MS type: Research article
Title: Aerosol Optical Properties Measurement using the Orbiting High Spectral Resolution Lidar onboard DQ-1 Satellite: Retrieval and Validation
Author: Chenxing Zha, Lingbing Bu, Zhi Li, Qin Wang, Ahmad Mubarak, Pasindu Liyanage, Jiqiao Liu, and Weibiao Chen

Dear editor and David Winker:
We deeply appreciate your valuable time in reviewing our manuscript again and providing us with constructive feedback and suggestions. We carefully considered each comment and made revisions to our manuscript. We sincerely hope that the revised manuscript can be published in Atmospheric Measurement Technology.

**General comments**

**I would like to thank the authors for making significant improvements to the first version of the manuscript. Several corrections and additional details are needed before publication though. Several of my comments below are related to the comparisons between DQ-1 and CALIPSO and the authors' speculation on reasons for differences between the retrievals.**

Responses to general comments: We are very thankful to you for your kind words and positive feedback about our manuscript. Based on your comments, we have made revisions to the manuscript, especially regarding the comparison between CALIPSO and DQ-1.

**Specific comments**

**Point 1: For aerosols, Equation 2.8 is a poor approximation of the correct relationship, which has a dependence on scattering ratio. The correct expression can be found in Tesche et al. JGR 2009 (Equation 11). Use of Eqn 2.8 will cause particulate depolarization to be underestimated for aerosols, because the scattering ratios are usually quite low. Cirrus have much higher scattering ratios and Eqn 2.8 can be a useful approximation.**

**The particulate depolarization of the dust layer in Fig 5e looks too small. Tesche et al. (2009) found a campaign-average particulate depolarization ratio of 30% for desert dust during the SAMUM campaign in the western Sahara desert. As mentioned above, use of Eq 2.8 to derive particulate depolarization from the measured volume depolarization underestimates the particulate depolarization.**

Response 1: Thank you for raising this question. Regarding the depolarization ratio, we have updated our retrieval method according to the paper you provided. The particulate depolarization ratio in the manuscript have been modified as:

Line 175:

The particulate depolarization ratio is expressed as:

$$\delta_p(r) = \frac{\beta_m(r)[\delta(r) - \delta_m(r)] + \beta_a(r)\delta(r)[1 + \delta_m(r)]}{\beta_m(r)[\delta_m(r) - \delta(r)] + \beta_a(r)[1 + \delta_m(r)]}, (Tesche\ et\ al., 2009) \qquad (2.8)$$

Line 258: The retrieval results of the particulate depolarization ratio from CALIPSO exhibit a mean value of 0.3, which is consistent with DQ-1.

Figure 5e:

[Figure]

Figure 5i:

[Figure]

**Point 2: Line 248 – there is a 400 km offset between the two instruments. The lack of cirrus at 15 km in Fig 5d is likely because there was no cirrus in the curtain observed by CALIOP. CALIOP can detect cirrus with extinction even less than 0.01 /km. I am confident the cirrus observed by DQ-1 at 15 km in Fig 5c would have been detected if CALIPSO had been flying in the same orbit as DQ-1.**

Response 2: Thank you for pointing out our negligence. The difference in the results of high-level cloud observations between the two systems is due to spatial differences. The corresponding modifications were made to the manuscript at:

Line 248: Due to spatial differences, the results of high-level clouds obtained by the two systems are different.

**Point 3: How was the CALIPSO depolarization profile in Figure 5i derived? The data in Fig 5f looks too noisy to produce the smooth profile in Fig 5i. My analysis of the CALIPSO particulate depolarization profile within the dust layer between 24N and 30N showed the profile of depolarization is quite noisy. I found the mean particulate depolarization was 25%.**

**Line 258 – when I look at this scene, I find some noise spikes giving particulate depolarization as high as 0.5 but mean values are less than 0.3.**

**Response 3:** Thank you for raising this issue. The inconsistent results in Figure 5i were due to excessive use of moving averages, and we no longer perform moving average on the CALIPSO profile in Figure 5i. The average value of the modified CALIPSO depolarization ratio is 0.3, which is consistent with the results of DQ-1. The corresponding modifications were made to the manuscript at:

Line 258: The retrieval results of the particulate depolarization ratio from CALIPSO exhibit a mean value of 0.3, which is consistent with DQ-1.

Fig 5i:

[Figure]

**Point 4: Lines 262-264 seem to attribute the difference in particulate depolarization seen in Fig 5e, 5f, and 5i to a difference in SNR between the two instruments. CALIOP and DQ-1 initially had similar 532 nm pulse energy (125 mJ). The pulse energy of the CALIOP laser decreased by 10% to 20% over the life of the mission, which decreased the SNR by 5-10%. This change in SNR would have only a small effect on retrievals and does not explain the differences seen here between DQ-1 and CALIOP data. In the standard CALIPSO data processing of the dust layer in this scene, CALIOP backscatter profile data is only averaged 1 km horizontally – much less than for DQ-1. This is likely the major reason that CALIOP data appears noisier. But the difference between the DQ-1 and CALIOP mean depolarization is probably due to the use of Eqn. 2.8 for DQ-1.**

**Response 4:** Thank you for raising this question. As mentioned in the Point 1, the depolarization ratio of the particles we previously retrieved was biased towards smaller values. After using a new retrieval method to calculate, the depolarization ratios of the two systems showed better consistency, and the signal quality of CALIPSO is sufficient for accurate retrieval. The corresponding modifications are the

same as the Response 1.

**Point 5: Table 1. It is not clear what 'Retrieval result error' refers to. Retrieval of which product? Under what conditions? Retrieval error usually depends on SNR so varies with altitude and is different for day and night. The text should describe how this error was determined - from validation inter-comparisons?**

Response 5: Thank you for raising this important issue and we are sorry for our negligence here. The inversion result error refers to a relative error of less than 15% when comparing the depolarization ratio and backscatter coefficient obtained from retrieval with other authoritative data products in low altitude below 6 km under nighttime conditions. The corresponding modifications were made to the manuscript at:

Table 1:

| Parameter | Value |
|---|---|
| Laser Wavelength | 532.245 nm |
| Laser Energy | ≥120 mj for pulses A and B |
| Laser Frequency Stability | 1MHz@10000s |
| Laser repetition frequency | 40 Hz |
| Telescope aperture | 1000 mm |
| Field of view | 0.2 mrad |
| Broadband bandpass filter | 0.45 nm |
| Narrowband FP filter | 30 pm |
| HSRL filters | iodine vapor filter, 1110 line |
| | aerosol signal suppression ratio ≥ 25 dB |
| Overall optical efficiency (excluding iodine filter) | 0.16 at parallel polarized channel |
| | 0.561 at perpendicular polarized channel |
| | 0.375 at high spectral resolution channel |
| Quantum efficiency of the detector | 40% |
| Retrieval result error | 15%* |

**\* The relative error between the DQ-1 retrieval results (backscatter coefficient and depolarization ratio) and other authoritative data products, at low altitudes below 6 km under nighttime conditions.**

**Point 6: It is mentioned in the author's Response to Reviewers that the HSRL retrieves a molecular depolarization of 1% at high altitudes, but this not mentioned in the manuscript. A broadband measurement of molecular depolarization should give a value somewhat larger than 1%. The DQ-1 HSRL uses a narrowband FP filter with a bandwidth of 30 pm. In this case the molecular depolarization should be about 0.3% because the filter rejects the Raman scattering lines and only passes the central Cabannes line, which is less depolarized. Do the authors know the polarization purity of the transmitted beam?**

Response 6: Thank you for raising this important question. In the Response letter (*https://doi.org/10.5194/amt-2023-219-AC3*), we did not consider the polarization purity of the transmitted beam when retrieving the depolarization ratio of high-altitude molecules. In fact, according

to the research on DQ-1 ACDL laser conducted by the Shanghai Institute of Optics and Fine Mechanics (SIOM), the polarization of the transmitted beam is 0.5% (Chen et al., 2023). After considering the polarization purity of the transmitted beam, the mean depolarization ratio of high-altitude molecules obtained by our retrieval is 0.5%, which is close to the theoretical value. The remaining bias is caused by the influence of the ACDL optical system. Furthermore, the analysis of the depolarization ratio of high-altitude molecules has been added to the manuscript at:

Line 125: To verify the accuracy of the parallel and perpendicular channels, we retrieved the depolarization ratio of atmospheric molecules at high altitude. The results, shown in Figure 1, indicate that the depolarization ratio is less than 0.5%, which confirms the accuracy of the two optical channels.

Figure 1:

[Figure]

Figure 1 Retrieval results of depolarization ratio at high altitude.

**Minor comments:**

**Point 7: Line 117 says the laser beam points off-zenith at 'a specific angle'. How far off zenith is the laser pointed and is the angle always the same, or has it changed during the DQ-1 mission?**
**Response 7:** We are sorry for our negligence here. The laser beam deviates from the zenith at an angle of 2 degrees and remains unchanged under the control of the attitude control system. The corresponding modification was made to the manuscript at:

Line 117: The laser beam is off-zenith, pointing at an angle of 2 degrees and remains steady due to the attitude control system.

**Point 8: Line 134-136: SNR is more difficult to estimate than signal magnitude, how is the threshold for 'SNR control' determined? Is the threshold actually defined in terms of SNR, or in terms of signal magnitude?**

**Response 8:** Thank you for raising this issue and we are sorry for our negligence. As you mentioned, SNR is difficult to estimate, so we use the magnitude of the weak echo signal beneath the dense cloud cover to determine the threshold. The corresponding modification was made to the manuscript at:

Line 136: The threshold is determined by the magnitude of the weak echo signal beneath the dense cloud cover.

**Point 9: Line 162-please define "S6 molecular model" or provide a reference**
**Response 9:** We are sorry for our negligence. The corresponding references have been added to the manuscript at:

Line 162: The molecular backscatter coefficient and extinction coefficient are calculated by the S6 molecular model (Tenti et al., 1974) using the data of temperature and pressure provided by ERA5.

**Point 10: Line 252-253 – the authors should be consistent in the units used for backscatter, choose either /km/sr or m/sr and use consistently throughout the paper.**
**Response 10:** Thank you for raising this important issue and we are sorry for our negligence in our manuscript. In the revised manuscript, the unit has been uniformly changed to km/sr and double checked.

**Minor corrections**

**Point 11:**
**Line 17 – 'prominently' is not the right word here; maybe 'clearly', or just say that retrieval algorithms and validation are necessary**
**Line 20 – 'showing' rather than 'describing'?**
**Line 80 – 'a' spaceborne HSRL sounds better here than 'the' spaceborne HSRL**
**Line 116 – 'The laser produces two …' rather than 'The laser is with two …'**
**Line 122 – 'high spectral resolution channels' rather than 'high spectral channels' ?**
**Line 207 – 'religions' should be 'regions'**
**Line 228 – stratocumulus clouds would not be found at an altitude of 15 km. These are likely some type of cirrus**
**Line 228 – rather than 'satellite's emitted laser', maybe 'laser return signal'**
**Response 11:** Thanks for pointing out these minor corrections in our manuscript. The corresponding modifications were made to the manuscript at:

Line 17: Developing a suitable retrieval algorithm and validating retrieved results are necessary.

Line 20: The results have shown a continuous profile alignment between the two datasets, with DQ-1 showing an improved signal-to-noise ratio (SNR).

Line 80: Furthermore, under the leadership of NASA, the Atmosphere Observing System (AOS) international program analyzes the additional value provided by a spaceborne HSRL system.

Line 116: The laser produces two distinct pulses, pulse A and pulse B, to observe the atmosphere practically, both of the pulses are normalized prior to the retrieval process.

Line 122: The high spectral resolution channels function to separate Mie scattering and Rayleigh scattering in the signal, obtaining the molecular scattering profile.

Line 207: With more than 70 well-established observational stations worldwide, MPLNET has several underlying surface conditions, allowing it to collect ongoing aerosol vertical profiles in different regions.

Line 228: At an altitude of 15 km, the distribution of cirrus was observed, with the laser return signal failing to penetrate certain portions of the cloud cover.

**Point 12: Line 20 – the meaning of 'continuous profile alinement' is not clear, does this mean the lidar profiles from the two instruments are similar (when the orbit tracks are close)?**
**Response 12:** We are sorry for our wording here. What we want to express here is that the profiles from the two instruments are similar. We made modification to the sentence as:

Line 20: The results indicate that the profiles of the two datasets are in good agreement, with DQ-1 showing an improved signal-to-noise ratio (SNR).

**Point 13: Lines 129-130 – Regarding "the filtered signal consists of no aerosol Mie scattering", I agree there is no significant aerosol signal visible in the filtered signal curve of Fig 1b but strong Mie scattering might result in significant leakage into the filtered signal. From the inset in Figure 1a, it looks like the unfiltered signal peak at 6 km altitude in Fig 1b will enhance the filtered Rayleigh signal by perhaps 20%.**
**Response 13:** Thank you for raising this issue and we are sorry for our mistake here. The removal of all Mie scattering signals is an idealized assumption. In practice, a small portion of Mie scattering signals will remain. As you mentioned, these Mie signals hold no significance in the filtered signal. The revised sentence is:

Lines 129-130: Figure 1b shows the comparison of signals before and after filtering, with no significant aerosol Mie scattering signal in the filtered signal, presenting a residual portion of molecular Rayleigh scattering.

**Point 14: Line 139 – Is 16.1 μsec the correct value? According to Dai et al. (2023), the time delay between pulses A and B is 200 μsec.**
**Response 14:** We apologize for our mistake. After double-checking, we found that the correct time delay is 200 μs. The sentence have been modified accordingly at:

Line 139: There is an energy difference between laser pulses A and B, where the L2A data have been calibrated during the production, and the time delay of pulses A and B is 200 μs.

**The above is the complete response to your comments. We look forward to hearing from you**

**regarding our responses. We would be glad to respond to any further questions and comments you may have.**

**Reference**

Chen, W., Liu, J., Hou, X., Zang, H., Ma, X., Wan, Y., and Zhu, X.: Lidar Technology for Atmosphere Environment Monitoring Satellite, Aerospace Shanghai (Chinese & English), 40, 13-20, 10.19328/j.cnki.2096-8655.2023.03.002, 2023.